# Experimental study of thermal rectification in suspended monolayer graphene

Haidong Wang[1,*], Shiqian Hu[2,3,4,*], Koji Takahashi[5,6], Xing Zhang[6,7], Hiroshi Takamatsu[1] & Jie Chen[2,3,4]

Thermal rectification is a fundamental phenomenon for active heat flow control. Significant thermal rectification is expected to exist in the asymmetric nanostructures, such as nanowires and thin films. As a one-atom-thick membrane, graphene has attracted much attention for realizing thermal rectification as shown by many molecular dynamics simulations. Here, we experimentally demonstrate thermal rectification in various asymmetric monolayer graphene nanostructures. A large thermal rectification factor of 26% is achieved in a defect-engineered monolayer graphene with nanopores on one side. A thermal rectification factor of 10% is achieved in a pristine monolayer graphene with nanoparticles deposited on one side or with a tapered width. The results indicate that the monolayer graphene has great potential to be used for designing high-performance thermal rectifiers for heat flow control and energy harvesting.

[1] Department of Mechanical Engineering, Kyushu University, Fukuoka 8190395, Japan. [2] Center for Phononics and Thermal Energy Science, School of Physics Science and Engineering, and Institute for Advanced Study, Tongji University, Shanghai 200092, China. [3] China–EU Joint Lab for Nanophononics, School of Physics Science and Engineering, Tongji University, Shanghai 200092, China. [4] Shanghai Key Laboratory of Special Artificial Microstructure Materials and Technology, School of Physics Science and Engineering, Tongji University, Shanghai 200092, China. [5] Department of Aeronautics and Astronautics, Kyushu University, Fukuoka 8190395, Japan. [6] International Institute for Carbon-Neutral Energy Research (WPI-I2CNER), Kyushu University, Fukuoka 8190395, Japan. [7] Department of Engineering Mechanics, Tsinghua University, Beijing 100084, China. * These authors contributed equally to this work. Correspondence and requests for materials should be addressed to X.Z. (email: x-zhang@tsinghua.edu.cn) or to H.T. (email: takamatsu@mech.kyushu-u.ac.jp) or to J.C. (email: jie@tongji.edu.cn).

Control and manipulation of heat transport has been a key goal for thermal engineering and energy utilization. One of the most attractive issues associated with thermal management in solid-state devices is thermal rectification[1–3]. Thermal rectification is a diode-like behaviour where the heat flux changes asymmetrically as the direction of the temperature gradient is reversed[4,5]. Due to its wide potential applications in stand-alone thermally driven computing systems[3] and efficient thermal energy conservation and storage systems[2], extensive attention has been given to this field in the past decade. For the bulk materials, researchers have pointed out that the thermal rectification occurs in a two-segment bar if the temperature dependence of thermal conductivity is different for each segment[6–10]. For the asymmetric nanostructures, more significant thermal rectification could be expected for different mechanisms[3]. As the thinnest two-dimensional material discovered so far, graphene is widely believed to be a promising candidate for achieving high thermal rectification factor[11–15]. Many molecular dynamics (MD) simulations have been reported for different asymmetric graphene nanostructures with thermal rectification factors $\eta = |\lambda_F - \lambda_B|/\lambda_B$ from 10–120% ($\lambda_F$ and $\lambda_B$ are the thermal conductivities in the forward and backward heat flow directions)[3,12–14,16–19].

However, as far as we know, thermal rectification in suspended monolayer graphene has not been realized experimentally. This article reports the monolayer graphene thermal rectifiers with different asymmetric designs. The experiments were conducted with five suspended monolayer graphene devices, including three asymmetric nanostructures: graphene with nanopores on one side, graphene with nanoparticles deposited on one side and graphene with a tapered width. In the first structure, the measured thermal rectification factor $\eta$ is $\sim 26\%$, while in the other two structures, $\eta$ is $\sim 10\%$. Theoretical and MD analysis show two different mechanisms for the varied thermal rectification behaviour in the different asymmetric graphene nanostructures.

## Results

**Fabrication of suspended graphene devices**. The monolayer graphene thermal rectifiers were prepared based on our previously reported method for suspending large-area graphene[20,21]. The sample quality was confirmed through Raman measurements (Supplementary Note 1 and Supplementary Fig. 1). Electron beam lithography and oxygen plasma etching can form the suspended graphene ribbon into any desired shape.

Figure 1 shows the structure of the H-type sensor and the five prepared monolayer graphene samples. With the large etching depth in the Si substrate ($>8\,\mu m$), a large-area monolayer graphene could be suspended and fabricated into a thermal rectifier. The first four samples were designed with a uniform width. Then, the asymmetric nanostructures were created in the graphene using a focused ion beam (FIB) or electron beam induced deposition as shown in Fig. 2. The fifth graphene sample was fabricated into a trapezoid shape. Thermal rectification was investigated in these asymmetric monolayer graphene structures.

The H-type sensor method was used to measure the thermal rectification effect in the experiments. This method meets the two basic requirements for examining thermal rectification. First, the heat flow direction through the graphene sample is reversible. Second, the thermal conductivity of graphene can be measured with high accuracy. As shown in Fig. 1, either sensor can be used as an electrical heater or thermometer. Thus, electrical power to each sensor can be adjusted to reverse the temperature gradient in the monolayer graphene ribbon. Reliable and valid thermal rectification can be confirmed only if the thermal rectification factor is much larger than the measurement error. H-type sensor offers a high-temperature resolution of 0.01 K. The estimated total measurement error was $<5\%$, which was much smaller than the observed thermal rectification factor (a detailed error analysis is provided in the Supplementary Note 2). The H-type sensor method was first developed to accurately measure the total hemispherical emissivity of a fine fibre[22]. In this case, even the smallest heat loss from the fibre to the environment through thermal radiation could be detected by the H-type sensor.

**Defect-engineered graphene thermal rectifiers**. The nanomanufacturing of the suspended monolayer graphene used a FEI Versa 3D™ dual-beam system, where the ion beam was used to create defects and the electron beam was used to observe the sample. The defect-engineered graphene samples are shown in Fig. 2.

Three monolayer graphene samples #1, #2 and #3 were modified by the FIB manufacturing (Supplementary Method 5 and Supplementary Figs 9–11). The sample #1 had 14 nanopores drilled on the bottom side. The average diameter of the nanopores was $\sim 100\,nm$. The sample #2 had 6 nanopores with an average diameter of $\sim 200\,nm$ while the sample #3 had 3 nanopores with an average diameter of $\sim 400\,nm$. The thermal conductivity before and after the defect engineering was measured in series in a high-vacuum chamber at $10^{-4}\,Pa$. In the experiments, one sensor was heated by a large direct current (DC) current, while the other sensor was used as a precise resistance thermometer to monitor the temperature rise caused by the heat conduction through the monolayer graphene. The thermal conductivity of graphene was then calculated based on a two-dimensional heat conduction model for the H-type sensor. The thermal analysis was performed using a commercial finite-element software COMSOL Multiphysics. Nine models were built in COMSOL (one model for sample #5 and two models for the other four samples) for thermal analysis of each graphene sample before and after nanomanufacturing. The contact thermal resistance, $R_c$, between the monolayer graphene and the sensor was analysed using a fin thermal resistance model[23]. The estimated $R_c$ was $\sim 12\%$ of the total thermal resistance of the pristine graphene and $\sim 3\%$ of that of the modified graphene (Supplementary Tables 1 and 2). More details on the thermal analysis can be found in the Supplementary Note 3. Since the thermal conductivity of graphene is temperature-dependent[20,24,25], the average temperature of the sample was kept the same in both heat flow directions. In this way, the change in the thermal conductivity by reversing the heat flux could be confirmed to be the result of the thermal rectification, not the temperature change.

Figure 3 shows the experimental results of graphene samples #1, #2 and #3. Before the defect engineering, there was no thermal rectification phenomenon observed in the pristine monolayer graphene. The solid and open symbols represent the thermal conductivities in both heat flow directions, which have a relative difference of $<2\%$. After drilling the nanopores on one side of the sample, the thermal conductivity in the direction from the nanopore region to the region without pores (solid symbols) was notably higher than that in the opposite direction (open symbols). The heat flux direction with larger thermal conductivity is indicated by the big red arrows in Fig. 2. The average thermal rectification factor $\eta_{ave}$ was $\sim 26\%$, much higher than the measurement error of 5%. A lattice dynamics model was used to calculate the thermal conductivity of monolayer graphene[26] with more details in the Supplementary Note 6.

The physical mechanism for the thermal rectification in the asymmetric monolayer graphene with nanopores is explained in Fig. 4. As mentioned above, the physical mechanism of the thermal rectification in bulk materials is that the thermal conductivity be a function of both space and temperature[6,7].

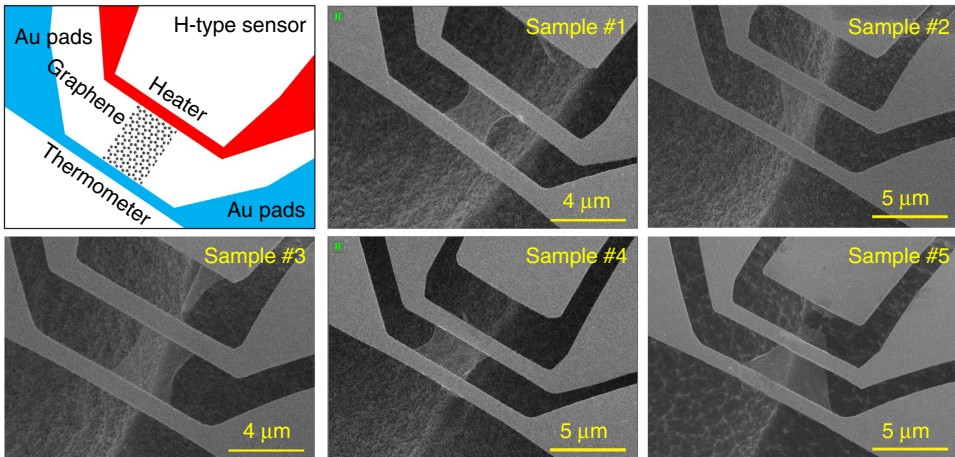

**Figure 1 | Scanning electron microscopy images of five suspended monolayer graphene samples.** The top-left figure shows the principle diagram of the H-type sensor. Two suspended Au micro-beam sensors were used as the heater and the thermometer with a graphene ribbon suspended between them. Five monolayer graphene samples were successfully fabricated for the experiments. The asymmetric structures are essential for realizing the thermal rectification in the monolayer graphene. In the first four samples, nanomanufacturing (FIB and electron beam induced deposition) was used to create different nanostructures on one side of the graphene as shown in Fig. 2. In the fifth sample, the graphene was designed with a tapered width.

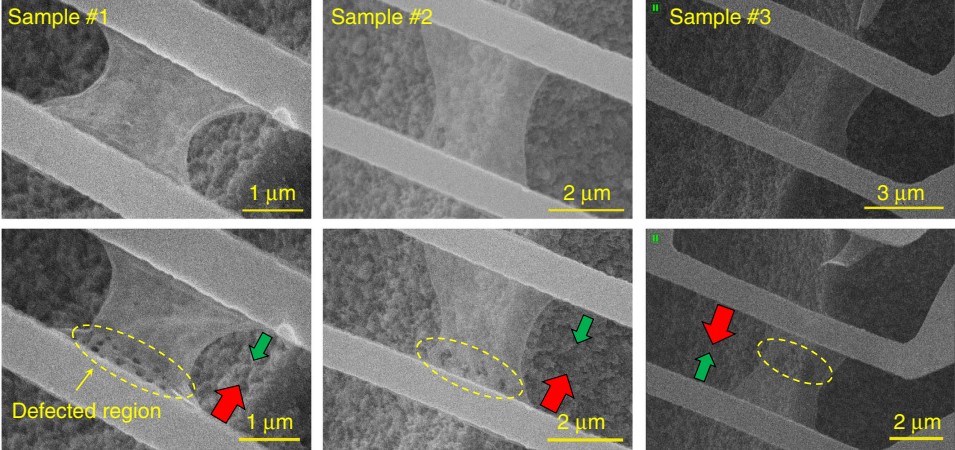

**Figure 2 | Scanning electron microscopy images of graphene before and after defect engineering.** Several nanopores were drilled in samples #1, #2 and #3 by the FIB system as seen in the yellow dashed line circles. The thermal conductivity of each modified graphene sample was then measured twice in opposite heat flow directions (shown by the red and green arrows). The big red arrows indicate the heat flow direction with the larger thermal conductivity, while the small green arrows indicate the opposite direction with the smaller thermal conductivity.

Similarly, we found that the dependence of the thermal conductivity on temperature and location is also the underlying reason for the thermal rectification in the asymmetric defect-engineered graphene samples[3]. After the FIB modifications, the thermal conductivity of graphene was significantly decreased from $\sim 2,100$ to $\sim 550\,\mathrm{W\,m^{-1}K^{-1}}$ as shown in Fig. 3. The temperature dependence of the thermal conductivity was also significantly different. For the pristine graphene, $\lambda_{\mathrm{P}}$ decreased with increasing temperature. For the modified graphene, the slope of the $\lambda$-$T$ line was much smaller. The modified graphene sample #2 even had an increasing $\lambda$-$T$ slope. On average, the $\lambda_{\mathrm{M}}$ of the modified graphene was almost independent of temperature. This experimental result is consistent with the recent MD simulation of defect-engineered graphene, where the exponent $\alpha$ for the temperature dependence $\lambda \sim T^{-\alpha}$ was found to decrease with the increase of defect ratio[27]. In pristine graphene, the temperature dependence of thermal conductivity is mainly due to the phonon Umklapp scattering, which increases with increasing temperature. After the defect engineering, the phonon-defect scattering

dominates the other scattering sources. The defect density was determined by only the ion beam dose and independent of temperature. As a result, $\lambda_{\mathrm{M}}$ became almost temperature independent. In Fig. 4, the red thick lines indicate the temperature-dependent thermal conductivity of the modified graphene, which can be calculated using a thermal resistance-in-series model as:

$$\lambda = \left[ \frac{L_1}{L_1 + L_2} \lambda_1^{-1} + \frac{L_2}{L_1 + L_2} \lambda_2^{-1} \right]^{-1}, \qquad (1)$$

where $L$ is the length and the subscripts 1 and 2 stand for the regions with and without nanopores. The thermal rectification factor of the modified graphene sample #1 is calculated as an example. Figure 2 indicates that $L_1/(L_1 + L_2) = 0.148$ for the sample #1. The estimated $\lambda_1$ of the region with nanopores is independent of temperature, which is $230\,\mathrm{W\,m^{-1}K^{-1}}$, $\sim 10\%$ of $\lambda_{\mathrm{P}}$, the thermal conductivity of pristine graphene[28]. Although the ion beam was focused on only the one region for drilling the

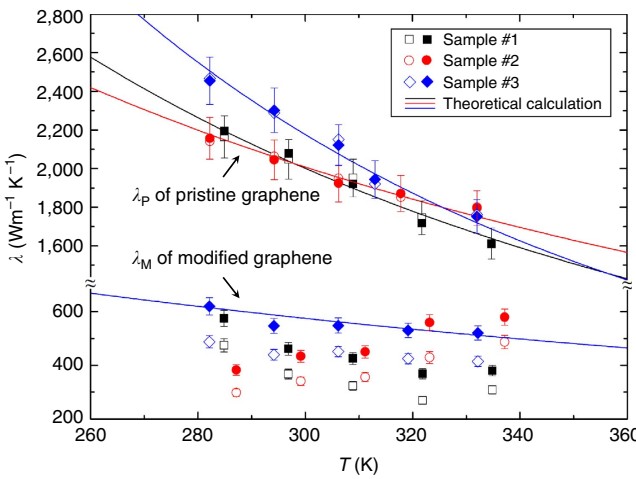

**Figure 3 | Thermal conductivities of the first three graphene samples.** For the pristine graphene before defect engineering, the measured thermal conductivity $\lambda_P$ was almost the same in two opposite heat flux directions (solid and open symbols). For the modified graphene with nanopores, the measured thermal conductivity $\lambda_M$ was larger in the direction from the nanopore region to the region without pores (solid symbols) than that in the opposite direction (open symbols). Here 5% error bars are plotted in the figure. A detailed uncertainty analysis is given in the Supplementary Note 2. The heat flux direction with larger thermal conductivity is indicated by the red arrows in Fig. 2. The average thermal rectification factor $\eta_{ave}$ of the modified graphene after defect engineering is 28, 26 and 25% for the samples #1, #2 and #3, respectively. The solid lines are the theoretical predictions based on a lattice dynamics model (in the Supplementary Note 6).

nanopores, some scattered ions might also damage the rest of the graphene and reduce the thermal conductivity throughout the sample. Thus, the thermal conductivity of the region without nanopore defects was assumed to be 24% of $\lambda_P$. $\lambda_2$ is temperature dependent, which is $380\,\mathrm{Wm^{-1}K^{-1}}$ in case I and $530\,\mathrm{Wm^{-1}K^{-1}}$ in case II of Fig. 4. Substituting all these parameters into equation (1) gives the total thermal conductivities $\lambda_{\mathrm{case\ I}}$ as $345\,\mathrm{Wm^{-1}K^{-1}}$ and $\lambda_{\mathrm{case\ II}}$ as $442\,\mathrm{Wm^{-1}K^{-1}}$. The calculated thermal rectification factor, $\eta$, is then 28%, which agrees well with the experimental result. It should be mentioned that the thermal rectification factor is dependent on the temperature difference between both sides of the graphene sample. If the temperature difference is small, the thermal rectification caused by the asymmetric distribution of thermal conductivity in graphene will be weakened.

**Graphene thermal rectifiers with asymmetric structures.** In addition to the defect-engineered samples, the asymmetric pristine graphene samples, #4 and #5, were also fabricated. The scanning electron microscopy images are shown in Fig. 5.

Graphene sample #4 had ten 200 nm diameter amorphous carbon nanoparticles deposited on one side. The nanoparticles can be clearly distinguished in the scanning electron microscopy image. For the sample #5, the graphene ribbon was designed with a tapered width. The asymmetric two-dimensional geometry can cause thermal rectification behaviour in the suspended graphene.

Figure 6 shows the measured thermal conductivities of samples #4 and #5 in the different directions. There was no thermal rectification observed in the sample #4 before the carbon nanoparticle deposition. After electron beam induced deposition process, the thermal conductivity of sample #4 was 10% higher

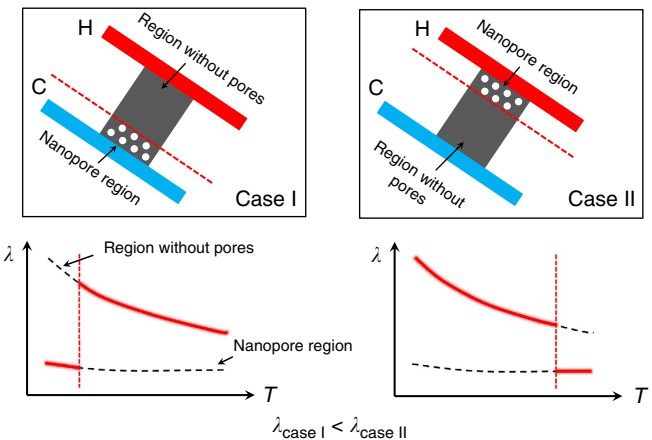

**Figure 4 | Explanation of the physical mechanism for the thermal rectification.** (Case I) Heat flows from the region without pores to the modified nanopore region. (Case II) The heat flow direction is reversed from the nanopore region to the region without pores, where the letters H and C indicate the hot and cold sensors. In the $\lambda$-$T$ curves, the black dashed lines represent the thermal conductivity of monolayer graphene. The higher line is the thermal conductivity of graphene without pores, while the lower line is the thermal conductivity of graphene with nanopores. The red thick lines that overlap the black dashed lines represent the temperature-dependent thermal conductivity of the asymmetric monolayer graphene with nanopores. The total thermal conductivity, $\lambda_{\mathrm{case\ II}}$, is larger than $\lambda_{\mathrm{case\ I}}$.

for heat flow from the clean region to the nanoparticle deposition region than in the opposite direction (indicated by the red arrow in Fig. 5). Similarly, the thermal conductivity of sample #5 was 11% higher for heat flow from the wide region to the narrow region than in the opposite direction. The observed thermal rectification in sample #5 corresponds with the 7% thermal rectification in a carbon nanotube thermal rectifier (higher thermal conductance was observed when the heat flowed from the wide region to the narrow region)[4]. It is noted that the samples #4 and #5 have notably higher thermal conductivities and much smaller thermal rectification factors than the other three defect-engineered samples. Moreover, the thermal conductivity decreases with increasing temperature in the last two samples, which is a typical sign for the good lattice quality. The observed temperature-dependent thermal conductivity in the last two samples suggest that the phonon-defect scattering is not the dominant phonon scattering mechanism, and the underlying thermal rectification mechanism should be different from that explained in Fig. 4. Moreover, it was reported that the boundary scattering in 2D graphene ribbon was not the same as that in 3D nanowires[29].

To understand the physical mechanisms of thermal rectification in sample #5, we have performed MD simulation and computed the spatial energy distribution for the propagating phonon modes in the non-equilibrium steady-state for the trapezoid graphene sheet (Methods section). The calculation result is shown in Fig. 7.

Figure 7 shows the calculated thermal rectification factor of trapezoid graphene and its spatial energy distribution for the propagating phonon modes. The MD simulation model is similar but much smaller than the experimental sample. For the smallest MD model, the size of the simulation domain is $L = 17\,\mathrm{nm}$, $W_1 = 22\,\mathrm{nm}$, $W_2 = 2\,\mathrm{nm}$ and the total number of C atoms is 7,420. We fix the angle $\theta$ in the trapezoid graphene and increase both $L$ and $W$ proportionally, extending the width $W_1$ to 200 nm, and 440 nm, respectively. There are 2,985,734 C atoms in the MD model for the largest width we considered. It is a formidable task

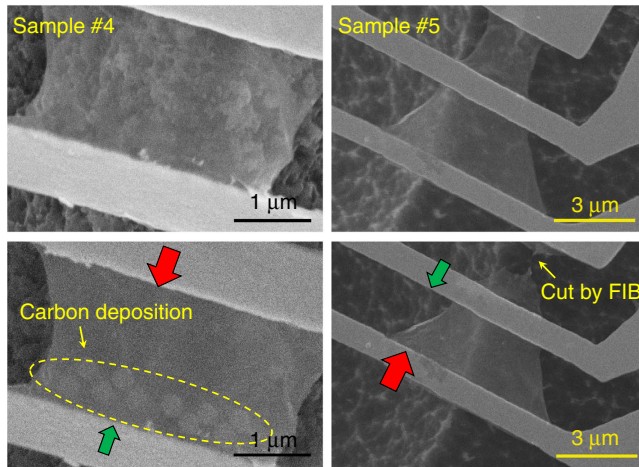

**Figure 5 | Scanning electron microscopy images of graphene with asymmetric carbon deposition or tapered width.** The images in the top row are the original suspended samples. The left-bottom image is the graphene sample #4 after amorphous carbon deposition by electron beam induced deposition. The deposited nanoparticles are marked by a yellow dashed-circle. The right-bottom image is the graphene sample #5 with a tapered width. The big red arrows indicate the heat flow direction with the larger thermal conductivity, while the small green arrows indicate the opposite direction with the smaller thermal conductivity.

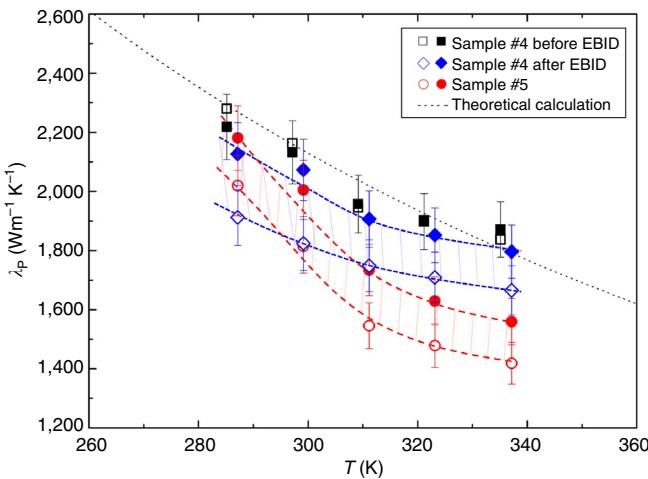

**Figure 6 | Thermal conductivities of the last two graphene samples.** The thermal conductivity of sample #4 was measured before and after depositing nanoparticles by electron beam induced deposition (EBID). Before EBID, the measured thermal conductivity of sample #4 was almost the same in two opposite heat flux directions (solid and open black symbols). After EBID, the thermal conductivity was 10% larger in the direction from the clean region to the nanoparticle deposition region (blue solid symbols) than that in the opposite direction (blue open symbols). For the sample #5, the thermal conductivity was 11% larger in the direction from the wide region to the narrow region (red solid symbols) than that in the opposite direction (red open symbols). Same as Fig. 3, 5% error bars are plotted in the figure. The heat flux direction with larger thermal conductivity is indicated by the red arrows in Fig. 5. The hatched areas highlight the difference in the thermal conductivity caused by the thermal rectification.

to further extend the MD simulation size to the micrometre scale, due to the tremendously increased computational load at large scale (see the Supplementary Note 7). The MD simulation result demonstrates that the thermal rectification ratio of trapezoid graphene decreases with increasing width (Fig. 7b), following a logarithmic curve. The simulation results agree well with the experimental data. Figure 7c,d indicates that the local energy $E$ of propagating modes is always smaller in the heat flux direction from the narrow end to the wide end. Here, we discuss the spatial energy distribution separately for the high and low temperatures. At the high temperature $T_h$, $E$ at the narrow end of graphene is smaller than that at the wide end, that is, $E_4 < E_1$. It reveals that the narrow end of graphene has stronger edge scattering effect on the propagating phonon modes than the wide end. Hence, the phonon mean free path and the local energy of propagating modes are smaller at the narrow end. At the low temperature $T_c$, $E$ at the wide end of graphene is smaller than that at the narrow end, that is, $E_3 < E_2$. This feature is associated with the low energy of propagating modes at the narrow end of graphene under the high-temperature condition. The phonon population density will be higher at the hot side of graphene than that at the cold side. If the hot side is the narrow end, the mean free paths of the large-population phonons will be reduced more. As a result, fewer propagating phonon modes can reach the cold side and the local energy at the cold side (wide end) is relatively small. In other words, the narrow end of graphene at high temperature appears to be the bottleneck for the propagating phonon modes, where the edge scattering affects a larger number of phonons than that at the wide end. In this way, the thermal conductivity in the heat flux direction from the narrow end to the wide end is smaller than that in the opposite direction.

Originated from the ultra-strong $sp^2$ bonding, graphene has unusually long phonon mean free path, which ranges from sub-micrometre to hundreds of micrometres at room temperature[30,31]. So the phonon mean free path of graphene is close to, or even larger than the sample size. In this case, the local energies of propagating modes at the two ends of graphene are coupled to each other. The propagating phonon modes at the cold side are

affected by the phonon scattering at the hot side. If the width or length of graphene is much larger than the phonon mean free path, the edge scattering becomes unimportant and the local energies at the hot and cold sides are independent to each other, then the thermal rectification disappears. This explains the declining rectification ratio of trapezoid graphene with the increasing width in the MD simulation (Fig. 7b).

Another approach to induce additional phonon scattering is via the mass loading to the graphene sheet, where the thermal conductivity can be suppressed. This is the case of graphene sample #4. To simulate the asymmetric structure of sample #4, we asymmetrically deposited heavy atoms on one side of the graphene in the MD model, as shown in Fig. 8.

The simulation result indicates that the heat flux in graphene is larger from the clean region to the other region with deposited heavy atoms, which is consistent with the experimental result of sample #4. We find that the deposited heavy atoms can cause significant phonon scattering in the graphene sheet. Similar to the narrow end in the trapezoid graphene sheet, the deposited region at high temperature appears to be the bottleneck for the propagating phonon modes in graphene, so that fewer propagating modes can reach the cold side. Hence, the local energy of propagating modes is smaller in the heat flux direction from the deposited region to the clean region, and the thermal conductivity is smaller as well. We have repeated the MD simulation by using different Lennard-Jones potential parameters and obtained similar results. More details about the MD simulation can be found in the Supplementary Note 7 and Supplementary Fig. 12. In the samples #1, #2 and #3, the phonon-defect scattering dominates over the other scattering mechanisms (Supplementary Note 6). The asymmetric dependence of thermal conductivity is the main reason for thermal rectification.

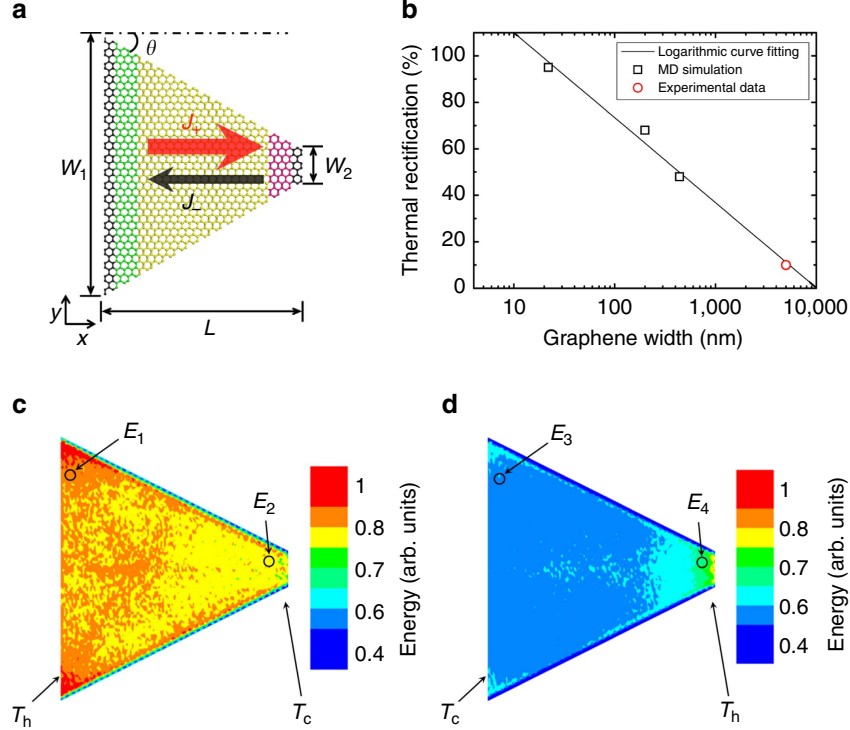

**Figure 7 | Thermal rectification of the trapezoid graphene analysed by MD simulation.** (**a**) Schematic picture of the trapezoid graphene. The angle $\theta = 30°$ is close to that of the graphene sample #5. The big red arrow (plus direction $J_+$) indicates the direction with higher thermal conductivity, while the opposite direction (minus direction $J$-) has lower thermal conductivity. (**b**) The thermal rectification ratio versus the graphene width $W_1$ (angle $\theta$ is constant). The logarithmic curve-fitting gives $y = -15.893 \ln(x) + 146.609$ with $R^2 = 0.989$. (**c**) Spatial energy distribution for the plus direction. (**d**) Spatial energy distribution for the minus direction. Here $T_h$ and $T_c$ denotes the high and low temperatures at two ends of graphene, and the spatial energy distribution includes those propagating phonon modes with the participation ratio larger than 0.4. The simulation results confirm that $E_4 < E_1$, $E_3 < E_2$, that is, local energy $E$ of propagating modes is always smaller in case (**d**) than that in case (**c**) at both high-temperature and low-temperature ends.

## Discussion

This work reports on two different thermal rectification phenomena in suspended monolayer graphene. The asymmetric defect-engineered samples had measured thermal conductivities ~26% higher for heat flow from the region with nanopores to the region without pores. Asymmetric pristine sample with tapered width/nanoparticles had measured thermal conductivity ~10% higher for the heat flow from the wide/clean region to the narrow/deposited region. The inseparable dependence of the thermal conductivity on the temperature and space is found to be the mechanism for the first thermal rectifier type. The asymmetric phonon scattering varying with the heat flux direction is found to be responsible for the second thermal rectifier type. The suspended monolayer graphene provides an excellent platform for designing two-dimensional thermal rectifiers which have potential applications in autonomous heat flow control and efficient energy harvesting.

## Methods

**Fabrication details.** We have developed a new method for preparing large-area suspended monolayer graphene. The graphene grown by chemical vapour deposition method was transferred onto $SiO_2$/Si substrate following a traditional polymethyl methacrylate (PMMA) method. Then the graphene was cut into micro ribbons by electron beam lithography and $O_2$ plasma etching. Different shapes and sizes of graphene ribbon can be designed in this step. After that, the metallic sensor and electrode pads were created on graphene by using an electron beam physical vapour deposition method. A graphene ribbon bridged between two metallic thin film sensors, forming an H shape. Then, another protection resist layer was spin-coated on the sample, leaving several uncovered windows for etching the substrate underneath. The $SiO_2$ layer and Si substrate were separately etched by buffered hydrofluoric acid and $XeF_2$ gas. Due to the isotropic nature of $XeF_2$ gas etching, the metallic sensors and graphene were completely suspended from the substrate. The

gas etching time was controlled by observing the suspended area under an optical microscope. Finally, the protection resist layer and $SiO_2$ layer at both sides of graphene were removed by wet etching. The whole device was dried following a standard supercritical point drying procedure to avoid rupture of graphene caused by liquid surface tension.

**Thermal conductivity measurement.** The suspended graphene ribbon provided the only heat conduction path between two metallic sensors. The temperature of each sensor was precisely controlled by adjusting the DC current heating power. The average temperature rises of both sensors were measured separately by using different 4-probe measurement suites (Supplementary Notes 2 and 5). All the measurements were carried out in a high-vacuum chamber of $10^{-4}$ Pa. In the steady-state, the thermal conductivity of graphene is directly related to the temperature difference between two sensors (Supplementary Figs 2 and 3). Assuming the same heat flux through the graphene ribbon, if thermal conductivity of graphene is higher, the temperature difference between two sensors is smaller. Input the measured temperature difference and geometrical sizes of graphene and sensor into a finite-element thermal analysis model, the thermal conductivity of graphene could be determined. More details about the thermal analysis model are provided in the Supplementary Note 4 and Supplementary Figs 4–8.

To measure the thermal rectification factor, the electrical heating power of each sensor was adjusted to reverse the heat flux direction. Then, the thermal conductivity of graphene was measured again following the same method. The measurement uncertainty analysis is provided in the Supplementary Note 2.

**Molecular dynamics simulations.** Non-equilibrium MD simulations were performed with the LAMMPS[32] package and the optimized Tersoff[33] potential for graphene to simulate the thermal rectification effect. Lennard-Jones potential was used to model the interaction between the randomly deposited nanoparticles and the graphene sheet. In our simulation, the time step was set as 0.5 fs. The fixed boundary condition was used in the x-direction, while the free boundary condition was used in the other two directions. To impose the temperature bias around the room temperature, two Nosé-Hoover thermostats[34] with different temperature are applied to the two ends of graphene. The thermal rectification ratio ($\eta$) was

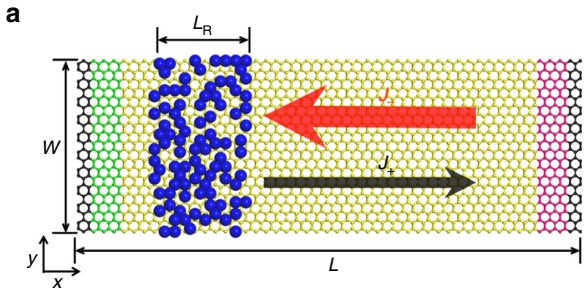

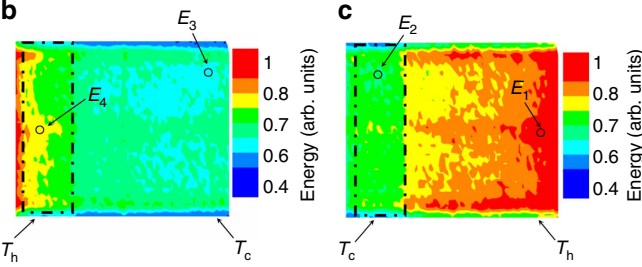

**Figure 8 | Spatial energy distribution of the asymmetrically deposited graphene.** (**a**) Schematic picture of the asymmetrically deposited graphene sheet. The blue atoms denote the deposited nanoparticles. We set in our simulation $L = 17$ nm, $W = 5$ nm, $L_R = 4$ nm and $d = 0.335$ nm. There are 400 heavy atoms deposited on the left side of the graphene sheet. The small black arrow indicates the plus direction $J_+$ with lower heat flux, while the big red arrow indicates the minus direction $J_-$ with higher heat flux. (**b**) Spatial energy distribution for the plus direction. (**c**) Spatial energy distribution for the minus direction. Here $T_h$ and $T_c$ denote the high and low temperatures at two ends of graphene, and the spatial energy distribution includes those propagating phonon modes with the participation ratio larger than 0.4. The dashed box in the figure denotes the deposited nanoparticles region. Similarly, the simulation results confirm that $E_4 < E_1$, $E_3 < E_2$, that is, the local energy $E$ of propagating modes is always smaller in case (**b**) than that in case (**c**) at both high-temperature and low-temperature ends.

computed as:

$$\eta = \left| \frac{J_+ - J_-}{J_-} \right|, \quad (2)$$

where $J_+$ and $J_-$ are the steady-state heat flux in two directions. More details about the simulation protocol can be found in our previous publication[35] and the Supplementary Note 7.

**Spatial energy distribution.** The spatial energy distribution on atom $i$ was defined as[36,37]:

$$E_i = \sum_\omega \sum_\lambda \sum_\alpha \left( n + \frac{1}{2} \right) \hbar \omega \varepsilon^*_{i\alpha,\lambda} \varepsilon_{i\alpha,\lambda} \delta(\omega - \omega_\lambda), \quad (3)$$

where $n$ is the phonon occupation number given by the Bose-Einstein distribution and $\varepsilon_{i\alpha,\lambda}$ is the $\alpha$th eigenvector component of eigen-mode $\lambda$ for the $i$th atom. To measure the energy distribution, the participation ratio ($P_\lambda$) for each phonon mode $\lambda$ was computed as[36–38]:

$$P_\lambda = \frac{1}{N \sum_i \left( \sum_\alpha \varepsilon^*_{i\alpha,\lambda} \varepsilon_{i\alpha,\lambda} \right)^2}, \quad (4)$$

where $N$ is the total number of atoms. To select the propagating phonon modes, only phonons with $P_\lambda > 0.4$ were included in the summation in Eq. 3. More details can be found in the Supplementary Note 7.

**Data availability.** The data that support the findings of this study are available from the corresponding author upon request.

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

## Acknowledgements

We thank Mr Tatsuya Ikuta, Dr Takanobu Fukunaga and Prof Yasuyuki Takata for their kind help in the micro/nano manufacturing and thermal measurement. The work was supported by the JSPS KAKENHI Grant-in-Aid for Young Scientists A (No. 17H04907) and National Natural Science Foundation of China Grant Nos. 51636002, 51327001, 51356001, 51506153 and 11334007. J.C. acknowledges support from the National Youth 1000 Talents Program in China, Shanghai Committee of Science and Technology in China (Grant No. 17ZR1448000) and the startup grant at Tongji University.

## Author contributions

H.T. and X.Z. proposed and supervised the project. H.W. designed and performed the experiments. S.H. and J.C. performed the MD simulations and prepared the discussion. K.T. helped to guide the fabrication of MEMS device. H.W. wrote the manuscript. H.T. and X.Z. checked and revised the manuscript.

## Additional information

**Competing interests:** The authors declare no competing financial interests.

