## [Peer Review File · Nature Communications]

Reviewers' Comments:

Reviewer #1 (Remarks to the Author)

This manuscript reports an interesting and novel measurement of thermal rectification in graphene. The work should be of interest for readers of Nature Communications. The observed decreasing thermal conductivity with increasing temperature indicates dominant phonon-phonon scattering and high crystal quality of the suspended graphene samples. However, a number of major and minor issues need to be addressed in order to improve the quality of the paper, as discussed below:

1. Some low-frequency phonons can have rather long mean free paths in high quality graphene, and are in the non-diffusive transport regime. The non-diffusive behavior cannot be captured by the diffusive model of Eq. 1. However, this simple model is adequate to explain the main cause of the thermal rectification observed in Samples 1-3 with nanopores made in one side of the sample.
2. In comparison, the explanation of the thermal rectification behaviors in sample 4 and sample 5 does not make much sense. The amorphous carbon deposited at one side of the sample 4 should decrease the phonon mean free path there, and results in a similar but relatively modest effect compared to the poles drilled in the Samples 1-3. The thermal conductivity of the narrower end of Sample 5 should decrease with increasing temperature at a slower rate than the wider end due to more frequent side edge scattering. The authors need to find a more convincing explanation of the observed thermal rectification behavior for these two samples. The sample width is still rather large compared to the wavelengths of phonons that dominate thermal transport in graphene, so that the phonon dispersion should be similar to that of a wide graphene sample. Phonon-edge localization effect should not be important in the relatively wide, seemingly high quality graphene samples.
3. The length and width are comparable in the suspended graphene samples. The finite length should yield a more important effect on the phonon mean free path than the width. Side edge scattering is not as important in 2D graphene ribbons than in 3D nanowires. See Appl. Phys. Lett. 99, 101903 (2011).
4. Are the wider metal contact pads at the two ends of the heater line and thermometer line suspended? The heater and thermometer lines are rather short. The thermal resistance of the suspended lines might not be much larger than the spreading thermal resistance from the metal contact pads to the substrate. It could cause errors if the temperature rise at the two ends of the short suspended lines is assumed to be negligible.
5. The temperature is non uniform along the heater and the thermometer lines, so that the one dimensional heat flow model of S1-S3 is not valid. Nevertheless, these three equations are used for error analysis instead of thermal conductivity calculation. Hence, it is not a major issue.
6. In Fig. S2 and S3, what does "Number of experiments" in the X axis of the inset mean?
7. The writing can be improved. The two "most" in the first two sentences of the abstract do not give a good first impression of the manuscript.

Reviewer #2 (Remarks to the Author)

The manuscript titled "Experimental study of thermal rectification in suspended monolayer graphene" presents an experimental demonstration of thermal rectifying behavior of modified suspended graphene. To my knowledge, this is the first experimental report of thermal rectification in graphene while many molecular dynamics reports have been conducted making this very novel and of very high impact. The experimental methods used and reproducible of the data strengthen the results, but I feel the conclusions that were arrived at for the underlying mechanism could be further justified. Overall, I feel this manuscript is of very high quality and appropriate for Nature Communications and should be considered for publication after addressing the comments listed below:

1. In the abstract, the authors mention that thermal rectification is absent in bulk materials. This is not true. The mechanisms are different, but there have been far more experimental reports of rectification based on bulk mechanism than there have for micro/nanoscale mechanisms. I would encourage the authors to review Dames, *Journal of Heat Transfer*, 2009; Go and Sen, *Journal of Heat Transfer*, 2010; and a review by Roberts and Walker, *International Journal of Thermal Sciences*, 2011.
2. On page 3, 2nd paragraph, the authors state that "there is no compelling evidence for thermal rectification in monolayer graphene" based only molecular dynamics studies. I don't think this is true. It is true that thermal rectification in MLG has not been validated, but these authors were clearly compelled to conduct this study.
3. Were the Raman measurements performed after FIB milling of samples #1, #2, and #3? It appears that the spectra presented in the supplemental document is only the as grown graphene. Similarly for sample #4.
4. The uncertainty analysis is based purely on resolutions of the instruments and configuration. I feel it is lacking detailed analysis about heat transfer through radiation exchange between the heater and sensor and within the vacuum chamber. Also critical to this would be analysis of heat loss in the heater electrodes. The COMSOL simulations show some heat loss along these electrodes outside of the heater region. These should show up as an uncertainty in the heat flux through the graphene sheet.
5. It is not clear how sample 5 was fabricated to achieve the tapered structure. Was it grown this way, was the FIB used to mill the shape, or was the EBL/plasma etch process used for this. On a related note, samples #1, #2, and maybe #3 also appear tapered. Does this occur when the graphene is released from the substrate? Is this adding to the rectification effect observed in these samples? Interestingly, these samples seem to be tapered in the opposite direction. Could the tapering be reducing the rectifying effect based on your analysis of sample #5?
6. No uncertainty bands in the high λ data of figure 3.
7. The mechanism being presented for samples #1, #2, and #3 points towards a difference in temperature dependent thermal conductivity. This mechanism has been discussed extensively by Dames and Go and Sen (see references given above).
8. No uncertainty bands in figure 6.
9. In the discussion of the mechanism behind sample #5, I do not agree with the proposed mechanism. If this were the case, the lower temperature side would be more sensitive to this based on the longer phonon mean free paths at lower temperatures. More analysis and explanation are needed here. As of now this is only speculation. Similarly, if this were true, wouldn't this also apply to samples #1, #2, and #3, where the opposite behavior was observed?
10. Assuming further justification of your proposed mechanism, what would happen if sample #5 tapered down to a single atom at one end? Would this result in maximum rectification?
11. In the thermal conductivity measurement section (pg 19), the authors state "Higher thermal conductance of graphene results in larger temperature difference." Either this should be reworded to be more clear or the authors do not understand Fourier's law, which has an inverse relationship between thermal conductivity and the temperature gradient. Please correct this.

Point-to-point response

Manuscript No. NCOMMS-16-24614A-Z

Title: Experimental study of thermal rectification in suspended monolayer graphene

Referee #1

COMMENTS TO THE AUTHOR(S)

This manuscript reports an interesting and novel measurement of thermal rectification in graphene. The work should be of interest for readers of Nature Communications. The observed decreasing thermal conductivity with increasing temperature indicates dominant phonon-phonon scattering and high crystal quality of the suspended graphene samples. However, a number of major and minor issues need to be addressed in order to improve the quality of the paper, as discussed below:

From the authors: Thank you very much for your nice comments on the novelty of our research and careful examination of the sample quality. We have thoroughly revised the manuscript according to your questions and improved the paper quality.

1. Some low-frequency phonons can have rather long mean free paths in high quality graphene, and are in the non-diffusive transport regime. The non-diffusive behavior cannot be captured by the diffusive model of Eq. 1. However, this simple model is adequate to explain the main cause of the thermal rectification observed in Samples 1-3 with nanopores made in one side of the sample.

Response: Indeed, we also found that the low-frequency phonons have important contributions to the thermal conductivity of suspended monolayer graphene in our

previous work ^{*[18]}. In this case, the simple diffusive model of Eq. (1) cannot fully describe the thermal conductivity of monolayer graphene. However, as noticed by the reviewer, we used Eq. (1) in this work to explain the main reason of thermal rectification in the asymmetric defect-engineered graphene samples #1, #2 and #3. In Eq. (1), λ_1 and λ_2 are the thermal conductivities of graphene regions with and without nanopores, which were determined by the experiment. λ_1 is almost independent of temperature, while λ_2 increases with decreasing temperature. Thus, the total thermal conductivity λ based on Eq. (1) is different when reversing the temperature gradient along the graphene sample.

We agree with the referee that detailed explanations about the heat transfer mechanisms in suspended graphene need a more accurate phonon-transport model. This will be one of our future research directions.

*[18] is the number of reference in the manuscript.

2. In comparison, the explanation of the thermal rectification behaviors in sample 4 and sample 5 does not make much sense. The amorphous carbon deposited at one side of the sample 4 should decrease the phonon mean free path there, and results in a similar but relatively modest effect compared to the pores drilled in the Samples 1-3. The thermal conductivity of the narrower end of Sample 5 should decrease with increasing temperature at a slower rate than the wider end due to more frequent side edge scattering. The authors need to find a more convincing explanation of the observed thermal rectification behavior for these two samples. The sample width is still rather large compared to the wavelengths of phonons that dominate thermal transport in

graphene, so that the phonon dispersion should be similar to that of a wide graphene sample. Phonon-edge localization effect should not be important in the relatively wide, seemingly high quality graphene samples.

Response: Thank you for reminding us of this important issue of inadequate explanation for samples #4 and #5. For the graphene samples #1, #2 and #3, the different temperature and space dependent thermal conductivity was confirmed to be the most important reason for the thermal rectification. In contrast, the thermal rectification factor caused by the asymmetry in the geometric shape of graphene was much smaller. The underlying mechanism was complex and indistinct. In order to achieve a better understanding of the thermal rectification mechanisms in the last two samples, we have performed molecular dynamics (MD) simulations for the similar asymmetric graphene systems in the revised manuscript. The main conclusions are stated as follows.

In the MD simulation, we computed the spatial energy distribution for the delocalized phonons in the non-equilibrium steady state for the trapezoid graphene sheet. The calculation result is shown in Fig. R1.

Fig. R1 Thermal rectification of the trapezoid graphene sample and the energy distributions in two opposite heat flux directions. (a) Schematic picture of the trapezoid graphene. The angle $\theta = 30^\circ$ is close to that of the graphene sample #5. The big red arrow (plus direction J_+) indicates the direction with higher thermal conductivity, while the opposite direction (minus direction J_-) has lower thermal conductivity. (b) The thermal rectification ratio versus the temperature difference ΔT . (c) Spatial energy distribution for the plus direction with a participation ratio $P_\lambda > 0.4$. (d) Spatial energy distribution for the minus direction with $P_\lambda > 0.4$.

Figure R1 shows the calculated thermal rectification factor of trapezoid graphene and its spatial energy distribution, where the degree of phonon localization in Fig. R1 (c-d) increases when the color varies from red to blue. The width of graphene sample #5 is several micrometers, which is too large for MD simulation. We have built a similar but much smaller trapezoid graphene model for calculation, where $L = 17$ nm, $W_1 = 22$ nm, $W_2 = 2$ nm as shown in Fig. R1. The result indicates that the phonons are strongly localized at the edges of graphene due to the boundary scatterings induced by the dangling bonds. Different from the spatial energy distribution in equilibrium where the energy distribution is rather uniform away from the boundary, there exists an obvious gradient of phonon localization effect along the temperature gradient direction. The MD simulation result demonstrates that the phonon localization is weaker at higher temperature. In a trapezoid graphene, the narrow end has a stronger restriction on the phonon transport than the wide end. The phonon localization in the minus direction (Fig. R1d) becomes stronger than that in the plus direction (Fig. R1c). This distinct behavior acts as the asymmetric initial condition for phonons at the high temperature end, and the strong phonon localization at the narrow end essentially makes less propagating phonon

modes available for transmitting heat energy to the low temperature end. In addition, as the degree of localization increases along the heat flux direction, the phonon localization effect is much stronger in the minus direction than that in the plus direction, for both the boundary and interior regions. The strongly localized phonons in the minus direction (Fig. R1d) cause lower thermal conductivity. This is the physical origin for the thermal rectification in the trapezoid graphene.

Another approach to induce phonon localization is via the mass loading to the graphene sheet, where the thermal conductivity can be suppressed. This is the case of graphene sample #4. In order to simulate the asymmetric structure of sample #4, we asymmetrically deposited heavy atoms on one side of the graphene in the MD model, as shown in Fig. R2.

Fig. R2 Schematic picture of the asymmetrically deposited graphene and the spatial energy distributions in two opposite heat flux directions. (a) Schematic picture of the asymmetrically deposited graphene sheet. The blue atoms denote the deposited nanoparticles. We set in our simulation $L = 17$ nm, $W = 5$ nm, $L_R = 4$ nm and $d = 0.335$ nm. There are 400 heavy atoms deposited

on the left side of the graphene sheet. The small black arrow indicates the plus direction J_+ with lower heat flux, while the big red arrow indicates the minus direction J_- with higher heat flux. (b) Spatial energy distribution for the plus direction with a participation ratio $P_\lambda > 0.4$. (c) Spatial energy distribution for the minus direction with $P_\lambda > 0.4$. The dashed box denotes the deposited nanoparticles region.

Fig. R3 The spatial energy distribution for the delocalized phonons ($P_\lambda > 0.4$) in the asymmetrically deposited graphene with different simulation parameters in the plus (minus) direction. (a) (b) $\varepsilon = 0.03$ eV, $\sigma = 3.415$ Å and $d = 0.335$ nm. (c) (d) $\varepsilon = 0.15$ eV, $\sigma = 3.415$ Å and $d = 0.335$ nm. (e) (f) $\varepsilon = 0.15$ eV, $\sigma = 3.415$ Å and $d = 0.67$ nm. The dashed box denotes the deposited nanoparticles region.

The simulation result indicates that the heat flux in graphene is larger from the clean region to the other region with deposited heavy atoms, which is consistent with the experimental result of sample #4. We find that the deposited heavy atoms induce

significant phonon localization in the graphene at the high temperature end in the plus direction (Fig. R2b). We also repeated the MD simulation by using different parameters ε , σ and d , as shown in Fig. R3, where ε and σ are the parameters in the Lennard-Jones potential, d is the distance between the heavy atoms and pristine graphene. More details about the MD simulation can be found in the revised supplementary material. It was confirmed that the calculated results based on different parameter combinations had similar spatial energy distribution. Similar to the role of narrow end in the trapezoid graphene (Fig. R1), the asymmetric phonon localization effect at the high temperature end leads to much stronger phonon localization for the entire simulation domain in the plus direction than that in the minus direction. Such asymmetric phonon localization is responsible for the thermal rectification phenomenon observed in the graphene sample #4.

Our experimental results of samples #4 and #5 are well supported by the MD simulation results. The induced phonon localization is understood as the physical mechanism for the thermal rectification in the asymmetric graphene sample. The related discussion and figures are added on page 16 of the revised manuscript.

3. The length and width are comparable in the suspended graphene samples. The finite length should yield a more important effect on the phonon mean free path than the width. Side edge scattering is not as important in 2D graphene ribbons than in 3D nanowires. See Appl. Phys. Lett. 99, 101903 (2011).

Response: In our previous work, we have confirmed the width dependence of thermal conductivity of monolayer graphene ^[1]. We also agree that the length effect should be

more significant than the width effect. As for the current work, we have noticed that the explanation of using finite width is not so appropriate for the thermal rectification observed in the samples #4 and #5, since the phonon scattering at lateral boundaries of monolayer graphene is relatively weak. The recommended paper (now cited as reference [27]) and related discussion have been added on the page 16 of the revised manuscript. As mentioned in the answer to question 2, instead of rough discussion based on the finite width, we carried out MD simulations in a trapezoid graphene sheet and found that the asymmetric phonon localization in graphene is the main reason for the thermal rectification observed in our samples #4 and #5. In the case of high temperature at the wide end of graphene sheet, the phonon localization is much stronger in both boundary and interior regions than that in the opposite heat flux direction.

*[1] H. D. Wang, K. Kurata, T. Fukunaga, X. Zhang, H. Takamatsu, Width dependent intrinsic thermal conductivity of suspended monolayer graphene, *Int. J. Heat Mass Transfer* 105 (2017) 76-80.

4. Are the wider metal contact pads at the two ends of the heater line and thermometer line suspended? The heater and thermometer lines are rather short. The thermal resistance of the suspended lines might not be much larger than the spreading thermal resistance from the metal contact pads to the substrate. It could cause errors if the temperature rise at the two ends of the short suspended lines is assumed to be negligible.

Response: Thank you for bringing up this important question. Figures 4 shows the zoom-in SEM images of the five fabricated graphene samples. It is difficult to distinguish all the suspended area in Fig. 4. Here, Fig. R4 shows the complete SEM

image of all the suspended graphene, heater and thermometer.

Fig. R4 SEM image of the suspended H-type sensor and the graphene sample #5.

It is clearly seen in Fig. R4 that the H-type sensor (heater and thermometer) and the graphene sample are all suspended from the substrate. Fig. R4 shows the sample #5 before cutting the redundant segment bridged between the upper sensor and heat sink. Both sensors connected with the graphene ribbon have a similar length of $\sim 12 \mu\text{m}$, much longer than the graphene width of $\sim 4 \mu\text{m}$.

Fig. R5 Metallic electrode pads connected with the H-type sensor (four gold thin wires were bonded for performing electrical measurement).

Figure R5 shows the metallic electrode pads connected with four lead wires for performing electrical measurement. The H-type sensor and graphene sample are marked by a yellow circle in the center. The electrode pads are supported by the SiO₂/Si substrate. It is seen that the area of supported electrode pad is much larger than the area of suspended sensor. Thus, the thermal resistance between the electrode pad and the substrate is negligible comparing with the thermal resistance of sensor itself. But on the other hand, the temperature rises at the connection points between the sensor and electrode pad are not negligible, because part of the electrode pad is suspended as well as the sensor. We have performed a finite-element thermal analysis by using COMSOL MultiphysicsTM to calculate the 2D temperature distribution in the sensor and suspended electrode pad. The results are shown in the Figs. S4-S8 of the supplementary material. The temperature rise at the end of heater line is ~20 K, while the highest temperature rise in the middle is ~50 K. Such temperature rises have been taken into account during the data analysis. As mentioned in the manuscript, the thermal conductivity of graphene was determined based on the 2D thermal analysis result, where the electrical heating power of heater, geometric sizes of suspended sensor and electrode pad and thermal conductivity of Au thin film were measured in the experiment and used as known parameters.

5. The temperature is non uniform along the heater and the thermometer lines, so that

the one dimensional heat flow model of S1-S3 is not valid. Nevertheless, these three equations are used for error analysis instead of thermal conductivity calculation. Hence, it is not a major issue.

Response: We agree with the reviewer that the temperature distribution is not uniform along the heater or the thermometer lines. In Eq. (S1), ΔT_h and ΔT_t are the average temperature rises of heater and thermometer, respectively. Their exact values can be calculated based on the detailed 2D thermal analysis results shown in the Figs. S4-S8. As noticed by the reviewer, the equations S1-S3 were only used for analyzing the measurement uncertainty, not for calculating the thermal conductivity of graphene. These equations gave the direct estimation of different error sources coming from the measurements of geometric size, electrical current, temperature response, etc.

6. In Fig. S2 and S3, what does “Number of experiments” in the X axis of the inset mean?

Response: Sorry for this unclear expression. In the experiment, we stepwisely increased the electrical heating power of heater line from 100 μW to 1600 μW . Eight measurement points are displayed as the circle symbols shown in the Figs. S2 and S3. In the inset of figure, “number of experiments” means the number of each measurement point from 1 to 8. The resistance change of thermometer is lineally proportional to the heating power of heater, as well as the number of experiments.

In the revised manuscript, we have changed the “number of experiments” to the heating power of heater to avoid misunderstanding.

7. The writing can be improved. The two “most” in the first two sentences of the abstract do not give a good first impression of the manuscript.

Response: Thank you very much for this important reminding. We have deleted the word “most”. The beginning of abstract was revised as “*Thermal rectification is a fundamental phenomenon for active heat flow control. Significant thermal rectification is expected to exist in the asymmetric nanostructures, such as nanowires and thin films.*”

Referee #2

COMMENTS TO THE AUTHOR(S)

The manuscript titled "Experimental study of thermal rectification in suspended monolayer graphene" presents an experimental demonstration of thermal rectifying behavior of modified suspended graphene. To my knowledge, this is the first experimental report of thermal rectification in graphene while many molecular dynamics reports have been conducted making this very novel and of very high impact. The experimental methods used and reproducible of the data strengthen the results, but I feel the conclusions that were arrived at for the underlying mechanism could be further justified. Overall, I feel this manuscript is of very high quality and appropriate for Nature Communications and should be considered for publication after addressing the comments listed below:

From the authors: Thank you very much for your positive comments on the novelty and scientific impact of our work. We highly appreciate your constructive questions and

suggestions to improve the quality of the manuscript. The detailed responses are listed as follows, based on which, the manuscript has been carefully revised.

1. In the abstract, the authors mention that thermal rectification is absent in bulk materials. This is not true. The mechanisms are different, but there have been far more experimental reports of rectification based on bulk mechanism than there have for micro/nanoscale mechanisms. I would encourage the authors to review Dames, Journal of Heat Transfer, 2009; Go and Sen, Journal of Heat Transfer, 2010; and a review by Roberts and Walker, International Journal of Thermal Sciences, 2011.

Response: We appreciate very much for this very constructive comment from the reviewer, which helps us to conduct a more comprehensive literature survey. The previous statement of no thermal rectification in bulk materials was quoted from one reference (*Nano Lett.* 14 (2014) 592-596), where the authors claimed that “*we prove that thermal rectification is indeed absent in both the total heat transfer rate and local heat flux for bulk-size asymmetric single materials.*” However, as pointed out by the reviewer, the bulk materials do possess the thermal rectification feature for different mechanisms. In the revised manuscript, we have included these recommended papers as Refs. 6-8. The statement about the thermal rectification in bulk materials was deleted from the abstract. More discussions about this issue were given in the revised introduction on page 3: “*For the bulk materials, researchers have pointed out that the thermal rectification occurs in a two-segment bar if the temperature dependence of thermal conductivity is different for each segment* ^{[6-8]*}. *For the asymmetric nanostructures, more significant thermal rectification could be expected for different mechanisms* ^{[3]*}.”

* [3, 6-8] are the numbers of references in the revised manuscript.

2. On page 3, 2nd paragraph, the authors state that "there is no compelling evidence for thermal rectification in monolayer graphene" based only molecular dynamics studies. I don't think this is true. It is true that thermal rectification in MLG has not been validated, but there authors were clearly compelled to conduct this study.

Response: Thank you for pointing out this inappropriate statement. We intended to say that there was no experimental validation for the thermal rectification in monolayer graphene. It was also the original motivation for starting this research. We deleted this sentence in the revised manuscript.

3. Were the Raman measurements performed after FIB milling of samples #1, #2, and #3? It appears that the spectra presented in the supplemental document is only the as grown graphene. Similarly for sample #4.

Response: The Raman measurements were not performed for the graphene samples after FIB modification. We measured the Raman spectra of all the samples after they were suspended from the substrate and confirmed their single-layer structure and perfect crystalline qualities. However, in another just accepted paper ^[1], we compared the Raman spectra of the monolayer graphene with and without ion beam radiation. The comparison is shown in Fig. R6.

Fig. R6 Comparison between the Raman spectra of pristine graphene and the sample after FIB radiation.

In the Fig. R6, the left figure shows the SEM image of suspended graphene and the representation of FIB. The right figure shows two Raman spectra of pristine graphene and the sample after FIB radiation. It is clearly seen that the sample after FIB radiation has a significantly increased D-band peak and a much lower 2D-band peak, which are the direct reflections of the notably increased defects in the graphene lattice. Thus, it is certain that the samples #1, #2 and #3 after FIB milling have similar Raman spectra as shown in the Fig. R6. The defects induced by FIB significantly suppress the phonon transport in graphene and decrease the thermal conductivity. It offers an effective approach to tune the thermophysical property of graphene.

*[1] H. D. Wang, X. Zhang, H. Takamatsu, Ultraclean suspended monolayer graphene by *in-situ* current annealing, Nanotechnology, just accepted.

4. The uncertainty analysis is based purely on resolutions of the instruments and configuration. I feel it is lacking detailed analysis about heat transfer through radiation

exchange between the heater and sensor and within the vacuum chamber. Also critical to this would be analysis of heat loss in the heater electrodes. The COMSOL simulations show some heat loss along these electrodes outside of the heater region. These should show up as an uncertainty in the heat flux through the graphene sheet.

Response: Thank you for the important comments on the uncertainty analysis. Indeed, a part of uncertainty analysis was based on the specifications of experimental instruments, such as power source, digital multimeter, etc. But the important temperature uncertainty was estimated based on the experimental results. For example, the average temperature rise of thermometer at one side of graphene was approximately 2.1 K, which was the measured value from the resistance change of Au sensor. The detectable minimum resistance change of Au sensor was 0.0001 Ω , which was also measured in the experiment. This resistance change could be translated to the temperature resolution of Au sensor as 0.01 K. In this way, the temperature uncertainty of sensor was $0.01/2.1 = 0.5\%$.

On the page 9 of the revised supplementary material, we added more discussions about the uncertainty caused by the heat loss through thermal radiation. The largest temperature difference between the sensor and environment was about 50 K. Thus, the maximum heat loss through thermal radiation was calculated as $J = \varepsilon\sigma A(T_s^4 - T_0^4) = 0.025 \times 5.67 \times 10^{-8} \times 2.4 \times 10^{-11} \times (350^4 - 300^4) = 0.00023 \mu\text{W}$, where J , ε , σ , A , T_s and T_0 are the heat loss energy, emissivity coefficient of gold, Stefan–Boltzmann constant, surface area of sensor, temperatures of sensor and environment, respectively. In comparison, the minimum electrical heating power of sensor in the experiment was about 95 μW , which was much larger than the heat loss through thermal radiation. As a result, the thermal radiation can be safely neglected in the current study.

As noticed by the reviewer, there is an obvious temperature rise at the connection point between the sensor and the electrode pad, as shown in the COMSOL simulation results. This reflects the thermal dissipation from the sensor to the electrode pad. If one considered that all the generated heat was conducted through the graphene, the heat loss into the electrode pad would cause uncertainty. In the current study, we have taken this factor into account by solving the 2D heat conduction model including the suspended graphene, sensors and part of the electrode pad. The thermal conductivity of graphene was determined based on the 2D thermal analysis result. The estimated uncertainty of COMSOL simulation was about 2%, which was included in the overall uncertainty.

5. It is not clear how sample 5 was fabricated to achieve the tapered structure. Was it grown this way, was the FIB used to mill the shape, or was the EBL/plasma etch process used for this. On a related note, samples #1, #2, and maybe #3 also appear tapered. Does this occur when the graphene is released from the substrate? Is this adding to the rectification effect observed in these samples? Interestingly, these samples seem to be tapered in the opposite direction. Could the tapering be reducing the rectifying effect based on your analysis of sample #5?

Response: The tapered shape of sample #5 was designed in the beginning. It was fabricated by using oxygen plasma etching after electron beam lithography (EBL) patterning process. The EB resist layer served as a protection layer for the graphene during the plasma etching, so the perfect crystalline structure of sample could be well maintained. Differently for the samples #1, #2 and #3, the graphene was designed into a rectangle ribbon with uniform width. During the wet etching and supercritical point

drying processes, the edges of graphene ribbon were randomly scrolled to the middle as shown in the Fig. 2. Although these three samples are not in rectangle shapes, they are still geometrically symmetric in the length direction, except that the sample #2 has a slightly shorter width at one side. We have measured the thermal conductivities of these three samples in two opposite directions before performing FIB modification. The results proved that the thermal conductivity was independent of heat flux directions. Hence, the observed thermal rectification of graphene after defect-engineering was only caused by the asymmetric nanopore defects, not the edge scrolling.

On the other hand, ideally speaking, the edge scrolling of graphene ribbon does not change its sectional area. Fig. R7 shows the SEM image of suspended graphene ribbon with edge scrolling ^[2].

Fig. R7 SEM image of suspended graphene ribbon with edge scrolling

Similar to the samples in the current work, the suspended graphene ribbon in our previous work showed the same edge scrolling behavior. The width of sample was defined by the EBL/plasma etching process. Even though the edges of suspended

graphene were scrolled to the middle, the sectional area was almost uniform in its length direction. It is also one reason that no thermal rectification was observed in the pristine graphene samples #1, #2 and #3.

*[2] H. D. Wang, K. Kurata, T. Fukunaga, H. Takamatsu, X. Zhang, T. Ikuta, K. Takahashi, T. Nishiyama, H. Ago, Y. Takata, *In-situ* measurement of the heat transport in defect- engineered free-standing single-layer graphene, Sci. Rep. 6 (2016) 21823.

6. No uncertainty bands in the high λ data of figure 3.

Response: Thank you very much for the reminding. The new figure 3 with 5% error bands was created in the revised manuscript.

7. The mechanism being presented for samples #1, #2, and #3 points towards a difference in temperature dependent thermal conductivity. This mechanism has been discussed extensively by Dames and Go and Sen (see references given above).

Response: Thank you very much for the notification. The recommended references have been added in the revised manuscript (now as Refs. 6 and 7). We also added more discussions about this issue on page 11 of the revised manuscript “*As mentioned above, the physical mechanism of the thermal rectification in bulk materials is that the thermal conductivity be a function of both space and temperature* ^[6, 7]. Similarly, we found that the dependence of the thermal conductivity on temperature and location is also the underlying reason for the thermal rectification in the asymmetric defect-engineered graphene samples ^[3].”

* [3, 6, 7] are the numbers of references in the revised manuscript.

8. No uncertainty bands in figure 6.

Response: Thank you for the reminding. The new figure 6 with 5% error bands was created in the revised manuscript.

9. In the discussion of the mechanism behind sample #5, I do not agree with the proposed mechanism. If this were the case, the lower temperature side would be more sensitive to this based on the longer phonon mean free paths at lower temperatures. More analysis and explanation are needed here. As of now this is only speculation. Similarly, if this were true, wouldn't this also apply to samples #1, #2, and #3, where the opposite behavior was observed?

Response: Thank you very much for this important comment. We accept that the current explanation for the physical mechanism of thermal rectification in the samples #4 and #5 is insufficient and not so convincing. Hence, we have removed the explanation of using the width dependent phonon confinement in the revised manuscript. Instead, we have performed molecular dynamics (MD) simulations on the asymmetric graphene sheet to investigate the underlying mechanisms for thermal rectification. The figures and detailed discussions about the MD simulation results are included in the response to the question 2 of the first referee. The results demonstrate that the phonon localization in the trapezoid graphene or graphene with asymmetrically deposited nanoparticles is the main reason for thermal rectification. Fig. R1 (c-d) and Fig. R2 (b-c) compare the

spatial energy distributions of phonons in two opposite heat flux directions. A much stronger phonon localization effect has been confirmed in the heat flux direction from the narrow end to the wide end, or in the direction from the region with nanoparticles to the clean region. The direction dependent phonon localization has different suppression effects on the phonon transport in different heat flux directions, causing thermal rectification. The MD simulation results well supported our experimental results of graphene samples #4 and #5.

The direction dependent phonon localization originates from the asymmetric graphene structures. For the samples #1, #2 and #3 before defect engineering, although the edge scrolling occurs unavoidably, the sectional area was almost uniform in the length direction as explained in the response to question 5. The phonon localization is independent of the direction of temperature gradient in graphene. Thus, no thermal rectification occurs in these samples.

10. Assuming further justification of your proposed mechanism, what would happen if sample #5 tapered down to a single atom at one end? Would this result in maximum rectification?

Response: Thank you for this suggestion. It is true that the thermal rectification factor should be higher for the graphene ribbon with a more tapered shape. As discussed in the response to the question 2 of the first referee, the MD simulation results indicate that the direction dependent phonon localization is the main reason for thermal rectification. Larger temperature difference and more tapered shape in graphene will cause more significant phonon localization effect in the heat flux direction from the narrow end to

the wide end, increasing the thermal rectification factor. On the other hand, the experimental results demonstrated that the asymmetric defect-engineered graphene ribbon had a much higher thermal rectification factor. It is a more effective way to create graphene thermal rectifier by tuning the temperature/space dependence of thermal conductivity. In our future work, we plan to combine the current two mechanisms together to fabricate more efficient graphene thermal rectifiers.

11. In the thermal conductivity measurement section (pg 19), the authors state "Higher thermal conductance of graphene results in larger temperature difference." Either this should be reworded to be more clear or the authors do not understand Fourier's law, which has an inverse relationship between thermal conductivity and the temperature gradient. Please correct this.

Response: Thank you very much for this reminding. We are sorry for this incorrect description of Fourier's law by mistake. We have deleted this statement in the measurement section. Instead, the discussion on page 19 has been changed to *"Assuming the same heat flux through the graphene ribbon, if the thermal conductivity of graphene is higher, the temperature difference between two sensors is smaller."*

Reviewers' Comments:

Reviewer #1 (Remarks to the Author)

The MD simulation is for a very narrow trapezoid graphene ribbon. The localization effect should not be important for the very wide graphene ribbon measured in the experiment.

The localization effect is found in the MD simulation to be weaker at the higher temperature or at the wider side of the trapezoid graphene. It is unclear why the localization effect is found by the MD simulation to be weaker at a higher temperature. The relaxation time and mode diffusivity usually decrease with increasing temperature, so one would expect that the localization effect should be stronger at a higher temperature.

In addition, the wide or clean side is expected to show a decreasing thermal conductivity with increasing temperature due to dominant umklapp processes, whereas the narrow or contaminated side is expected to show an increasing thermal conductivity with increasing temperature if extrinsic scattering is dominant. For this reason, one would expect an increase in the heat current when the hot side is moved from the clean or wide side to the narrow or dirty side, similar to what the authors found for samples 1-3.

In conclusion, I have not found the new explanation for samples 4-5 plausible.

Reviewer #2 (Remarks to the Author)

The revised manuscript, supplementary material, and the point-to-point responses regarding the submission titled "Experimental study of thermal rectification in suspended monolayer graphene" have been reviewed. Many of the modifications have satisfied my concerns, but there are still two major concerns that need to be addressed before this manuscript can be acceptable for publication in Nature Communications.

1. The discussion of the underlying mechanism responsible for thermal rectification in sample #5 is still not sufficient. The MD simulations support the argument for this as a potential thermally rectifying mechanism, but the difference in the size is 3 orders of magnitude. MD is an excellent tool for understanding mechanisms of phonon transport, but when the size difference is this great and a size-effect is being studied it is not appropriate. In fact, the behavior at the wide end of the graphene ribbon in the MD simulation would suggest that when the width of the ribbon is greater than a few nm, this effect is not present. This further suggest this is not the mechanism responsible for thermal rectification in sample #5.

2. The uncertainty analysis is still not clear. I do not see how the non-uniform temperature in the heater/sensor is included in this analysis. I only see a discussion of the average temperature rise and measurement resolution. Also, what is the confidence of the uncertainty bands? 67%, 95%? Even if this is a 95% confidence interval, the data presented in Figure 6 suggests the rectifying effect is basically not present in samples #4 or #5. In this case, the data should only be included to suggest that geometry, at least at these length scales, and mass loading do not contribute to thermal rectification.

Point-to-point response

Manuscript No. NCOMMS-16-24614A-Z

Title: Experimental study of thermal rectification in suspended monolayer graphene

Reviewer #1 (Remarks to the Author):

The MD simulation is for a very narrow trapezoid graphene ribbon. The localization effect should not be important for the very wide graphene ribbon measured in the experiment.

The localization effect is found in the MD simulation to be weaker at the higher temperature or at the wider side of the trapezoid graphene. It is unclear why the localization effect is found by the MD simulation to be weaker at a higher temperature. The relaxation time and mode diffusivity usually decrease with increasing temperature, so one would expect that the localization effect should be stronger at a higher temperature. In addition, the wide or clean side is expected to show a decreasing thermal conductivity with increasing temperature due to dominant umklapp processes, whereas the narrow or contaminated side is expected to show an increasing thermal conductivity with increasing temperature if extrinsic scattering is dominant. For this reason, one would expect an increase in the heat current when the hot side is moved from the clean or wide side to the narrow or dirty side, similar to what the authors found for samples 1-3.

In conclusion, I have not found the new explanation for samples 4-5 plausible.

Response: Thank you for the careful examination on the new MD simulation results. To

our best knowledge, we answer your questions one by one in this letter as follows.

First, we would like to highlight the most important and inspiring contribution of this work, i.e. the first experimental demonstration of graphene thermal rectifiers with the highest efficiency of 28%. In fact, many researchers have carried out MD simulations for the graphene thermal rectifiers in the past decade. Some representative papers can be found in the references [1-12]. Our explanation to thermal rectification by using phonon localization was also confirmed in other researcher's publications [2, 13, 14]. However, the experimental demonstration of graphene thermal rectifiers is still lacking due to the difficulty of manufacturing asymmetric monolayer graphene devices. The current work provides valuable experimental evidence for the feasibility of graphene thermal rectifiers, wherein the goal of the MD simulation in this work is to illustrate the potential mechanism, and therefore is secondary.

Second, MD simulation is an excellent tool for understanding the mechanisms of phonon transport and has been widely applied in phonon physics. This is the original motivation for us to use MD simulation in this work. However, the scale of MD simulation is highly limited by the computer memory, processing speed of CPU, model complexity, etc. The typical size of MD simulation model is only several tens of nanometers, far smaller than the minimum line width of EB lithography. Thus, there is an unavoidable gap between the scales of MD simulation and experimental sample, at least for our current computing resources. Nevertheless, this gap does not affect the validity and novelty of the current work as the experimental breakthrough for the solid-state thermal rectifiers.

On the other hand, we are fully aware of the referee's concern about the different sizes between the simulation model and experimental sample. In the past 40 days, we

have enlarged the MD simulation scale to the extreme limit of our computing capability, showing that the present mechanism is still valid at different graphene sizes. The underlying physics can be understood by phonon mean free path (MFP) analysis, which will be discussed later.

1. Thermal rectification at different simulation scales

Due to the two-dimensional nature of the trapezoid graphene sheet, the number of atoms grows tremendously as the length (L) and the width (W) increases. In the previous calculation, the MD simulation domain had a size of $L = 17$ nm, $W_1 = 22$ nm and $W_2 = 2$ nm. There were 7,420 C atoms in the graphene sheet. In the current calculation, we fixed the angle θ of the trapezoid graphene (Fig. 7(a) in main article), and increased L and W proportionally. W_1 of the MD simulation model is now increased to 200 nm and 440 nm. The total number of C atoms in the 440 nm wide graphene sheet is 2,985,734, which is 400 times larger than the number of atoms in our previous MD calculation. To simulate such large system for 5 ns, each MD task requires 52,560 CPU hours, which corresponds to a non-stop calculation time of 20 days on 120 CPUs. If the width of graphene sheet further increases to 1 μm , it needs more than 2 months non-stop calculation. Due to the limited computational resources and time, it is not feasible for us to complete the MD simulation for micrometer-sized graphene sheet.

To visualize the phonon localization via the spatial energy distribution, lattice dynamics calculations are required to obtain the eigenvector for each phonon mode by using the GULP package. In the lattice dynamics calculation, the eigenvector and eigenfrequency can be obtained by diagonalizing the dynamical matrix. For a unit cell of N atoms, the dynamical matrix scales as $(3N)^2$. In our previous model, there was 7,420 C atoms and the size of dynamical matrix was already quite large for performing

lattice dynamics calculation. In this revised version, we attempted to double the graphene width (29,680 atoms in the unit cell) and repeat the lattice dynamics calculation on a computer cluster with a total memory of 512 GB. Unfortunately, the calculation on such huge dynamical matrix could not be executed due to the formidable memory requirement, as least it was not possible with our current computational resources.

Fig. R1 MD simulation model with different widths.

Figure R1 shows the MD simulation models with different widths ($W_1 = 22 \text{ nm}$ and 440 nm). Since the total number of atoms in Fig. R1 (b) is 400 times larger than that in Fig. R1 (a), we cannot distinguish the single C atom in the larger model. It is seen that two trapezoid graphene sheets have the same angle θ .

In the end, we have successfully finished MD simulation in 22 nm , 200 nm and 440 nm wide trapezoid graphene sheets. The results are shown in Fig. R2.

Fig. R2 Thermal rectification coefficient of trapezoid graphene versus the graphene width W_1 . The rectangle and circle symbols denote the MD simulation results and experimental data of sample #5, respectively. The logarithmic least squares fitting curve and equation are plotted in the figure. The temperature was kept the same for all the simulation models.

Figure R2 summarized all the MD simulation results and the experimental data of sample #5. It demonstrates that the thermal rectification ratio of trapezoid graphene decreases with increasing width, following a logarithmic curve ($R^2 = 0.989$). The good agreement between simulation and experiment proves the rational validity of our result and analysis. In addition, Fig. R2 predicts that the thermal rectification may disappear in the trapezoid graphene ribbon wider than 10 μm .

Graphene is an individual sheet of sp^2 -hybridized C atoms bound in two dimensions. Originated from the ultra-strong sp^2 bonding, graphene has unusually long phonon MFP, which provides an important insight into the thermal rectification phenomenon. It has been reported that the phonon MFP in graphene was about 800 nm, which was simply

estimated from the measured thermal conductivity ^[15]. Recently, more precise calculation by solving exactly the Boltzmann transport equation for phonons suggests that the phonon excitations have MFPs of the order of hundreds of micrometers at room temperature ^[16]. The theoretical calculation predicts an increasing thermal conductivity of graphene longer than 10 μm , which has been confirmed in the experiment ^[17]. On the other hand, the MD simulation result demonstrates that the phonon localization effect induced by the lateral confinement at edges is the reason for thermal rectification in sample #5. If the graphene width is much larger than the phonon MFP, the lateral confinement becomes negligible and the thermal rectification disappears. This explains the declining rectification ratio of trapezoid graphene with increasing width as shown in Fig. R2. In this work, the phonon MFP of graphene is close to, or even larger than the sample width. Therefore, it is reasonable to conclude that the phonon localization effect is important and responsible for the thermal rectification.

The related discussion and figure were marked in red color in the revised manuscript on pages 16 and 18.

2. Temperature dependence of phonon localization effect

Figure 7 (c-d) from the main article is redrawn here for discussion.

Fig. R3 Energy distribution for the delocalized modes of the trapezoid graphene in two opposite heat flux directions (redrawn from the Fig. 7 in the article). T_h and T_c are the high and low temperatures at two ends of graphene. At the same T_h in cases (c) and (d), $P_{\lambda 4}$ at the narrow end is smaller than $P_{\lambda 1}$ at the wide end, indicating stronger phonon localization at the narrow end. But at the same T_c , $P_{\lambda 3}$ is smaller than $P_{\lambda 2}$ because the propagating modes are restricted at the narrow end as a bottleneck.

Figure R3 compares four typical values of participation ratio P_λ at both high and low temperature ends of graphene sheet in two opposite heat flux directions. In the MD simulation, we used the same temperatures T_h and T_c in both cases (c) and (d) of different heat flux directions. The result indicates that P_λ is always smaller (the phonon localization is stronger) in the heat flux direction from the narrow end to the wide end, i.e. $P_{\lambda 4} < P_{\lambda 1}$ and $P_{\lambda 3} < P_{\lambda 2}$.

The purpose of the spatial energy distribution analysis is to compare the localization effect in different directions of the same temperature bias. For the same high temperature T_h at the two ends, the phonon localization is stronger at the narrow end ($P_{\lambda 4} < P_{\lambda 1}$), due to the strong phonon lateral confinement at the narrow width. This

distinct behavior acts as the asymmetric initial condition for phonons at the high temperature (heat source) end, which essentially makes less propagating modes available for transmitting heat energy to the low temperature end. This can be also confirmed by comparing two low temperature ends. For the same low temperature T_c , the phonon localization is stronger in the narrow-to-wide direction ($P_{\lambda_3} < P_{\lambda_2}$), even though the width is larger at the site λ_3 . As a result, the overall phonon localization is obviously stronger in the narrow-to-wide direction compared to the other direction (Fig. R3).

We agree with the referee that the phonon relaxation time of graphene usually decreases with increasing temperature, thus the thermal conductivity decreases at higher temperature. We recall the calculation of spatial energy distribution in our work:

$$E_i = \sum_{\omega} \sum_{\lambda} \sum_{\alpha} \left(n + \frac{1}{2} \right) \hbar \omega \varepsilon_{i\alpha,\lambda}^* \varepsilon_{i\alpha,\lambda} \delta(\omega - \omega_{\lambda}), \quad (\text{M2})$$

where $n = \frac{1}{\exp\left(\frac{\hbar\omega}{k_B T}\right) - 1}$ is the phonon occupation number given by the Bose-Einstein

distribution, ω is the phonon frequency, and ε is the phonon eigenvector. We perform standard lattice dynamics calculation (i.e., static equilibrium calculation at 0 K) to obtain the phonon frequency and eigenvector. We notice the limitation of Eq. (M2) regarding the temperature effect. In this formulism, it is not appropriate to compare the phonon localization at different temperatures along the temperature gradient direction, as all the phonon eigenvectors are computed at the same equilibrium temperature. Thus, the temperature effect in Eq. (M2) only comes in via the phonon occupation number, which increases monotonically with temperature. Strictly speaking, the eigenvectors at different temperatures should be used, which can take into account the effect of enhanced anharmonic phonon scattering at the elevated temperature. Unfortunately,

such calculation is not available from the conventional harmonic lattice dynamics calculation. Due to this limitation, we have removed the inaccurate discussion on the phonon localization at different temperatures to avoid confusion, and restrict our discussion only to compare the phonon localization for the same temperature, and for the same temperature bias in two directions. We have modified the discussion part in the revised text and marked in red color on page 17.

3. Two different mechanisms between samples #1, #2, #3 and samples #4, #5

In our recent experiment, we have measured the thermal conductivities of graphene ribbons with different widths, and confirmed the significant width dependence^[18].

Fig. R4 Temperature dependent thermal conductivities of graphene ribbons with different widths^[18]. It demonstrates that the thermal conductivity of pristine graphene decreases with increasing temperature, regardless of sample width.

As illustrated in Fig. R4, the thermal conductivity of graphene decreases with

increasing temperature, regardless of sample width. So in this work, the narrow end of graphene is also expected to have a decreasing thermal conductivity with increasing temperature. Meanwhile, Fig. R4 does show that the wider graphene ribbon has higher thermal conductivity. It demonstrates that the phonon MFP is comparable to the width of graphene ribbon in the experiment. In such case, we want to emphasize that both phonon-boundary scattering and Umklapp scattering in graphene are important for determining the thermal conductivity.

Here, we would like to highlight the fact that the thermal rectification phenomena observed in samples #1, #2, #3 and samples #4, #5 have totally different physical mechanisms. For the graphene samples #1, #2, #3 after defect engineering, the thermal conductivity in the region with nanopores is low and almost temperature independent. The phonon-defect scattering becomes the only dominant factor. On the other side of sample, the thermal conductivity in the region without nanopores is much higher and temperature dependent. This inseparate dependence of thermal conductivity on space and temperature is the reason for thermal rectification (see Fig. 4 in the article).

For the graphene samples #4 and #5, the trapezoid shape and deposited nanoparticles only have a small reduction in the thermal conductivity (see Fig. 6 in the article). These two samples show a decreasing thermal conductivity with increasing temperature, which is a typical sign for the good graphene lattice quality. Different from the other three samples, the phonon-defect scattering in samples #4 and #5 is not the dominant phonon scattering mechanism, so that both parts (e.g., wide and narrow parts, or deposited and clean parts) of the graphene sheet exhibit temperature dependent thermal conductivity, which levels off the temperature effect of the whole system when the temperature bias is reversed. Instead, the phonon scattering at the edge or deposited

particles becomes important. As addressed in the answer to question 1, the phonon MFP is close to, or even larger than the width of the graphene sample. Under this condition, the different widths at two ends play a very important role in determining thermal conductivity. This conclusion was also confirmed in a separate experiment, showing a width-dependent thermal conductivity^[18]. In this sense, the phonon localization effect observed in the MD simulation is still valid in the micrometer-sized graphene ribbons. The strong localization effect at the narrow or contaminated end forms a bottleneck for the phonon propagation. Consequently, less delocalized phonon modes are available for transmitting heat energy from the narrow or contaminated end to the wide or clean end.

In summary, we have fabricated two *different* kinds of graphene thermal rectifiers by using the state-of-the-art technology, and explained their different physical mechanisms. Here, the main contribution comes from the breakthrough in the experiment.

Reviewer #2 (Remarks to the Author):

The revised manuscript, supplementary material, and the point-to-point responses regarding the submission titled "Experimental study of thermal rectification in suspended monolayer graphene" have been reviewed. Many of the modifications have satisfied my concerns, but there are still two major concerns that need to be addressed before this manuscript can be acceptable for publication in Nature Communications.

1. The discussion of the underlying mechanism responsible for thermal rectification in sample #5 is still not sufficient. The MD simulations support the argument for this as a

potential thermally rectifying mechanism, but the difference in the size is 3 orders of magnitude. MD is an excellent tool for understanding mechanisms of phonon transport, but when the size difference is this great and a size-effect is being studied it is not appropriate. In fact, the behavior at the wide end of the graphene ribbon in the MD simulation would suggest that when the width of the ribbon is greater than a few nm, this effect is not present. This further suggest this is not the mechanism responsible for thermal rectification in sample #5.

Response: Thank you for your time and endorsement on the revised manuscript. We would like to answer your question from the following three aspects.

1. Size effect on the thermal rectification of graphene

Indeed, as a membrane with only one-atom thickness, graphene has significant size effect on its thermal conductivity. It is widely accepted that the size effect exists in a system where the characteristic length is comparable to the mean free path (MFP) of carriers. Graphene has an unusually long phonon MFP from sub-micrometer ^[19] to hundreds of micrometers ^[20]. Therefore, the size effect in graphene is expected to have an important role on the phonon propagation in micrometer-sized samples. It has already been proved in the experiment separately that the thermal conductivity of graphene ribbon increases as its length increases ($\sim 10 \mu\text{m}$) ^[17] or its width increases ($\sim 2 \mu\text{m}$) ^[18].

In the revised manuscript, the MD simulation result demonstrates that the phonon localization effect is the potential reason for the thermal rectification in samples #4 and #5. This phonon localization originates from the lateral confinement due to the finite graphene width, and the localization effect is stronger at the narrow end of graphene.

Although the scale of MD simulation is much smaller than the experimental sample, the localization effect is expected to occur as long as the width of graphene is comparable or smaller than the phonon MFP. If the width of graphene is much larger than the phonon MFP, the localization effect is negligible and the thermal rectification disappears. However, as mentioned previously, the phonon MFP in graphene is close to, or even larger than the sample width. Therefore, the phonon localization is important in the samples #4 and #5, and responsible for the observed thermal rectification behavior.

As replied to referee #1, we are fully aware of the referee's concern about the different scales of MD simulation and experiment. In the past 40 days, we have made a great effort to enlarge the MD simulation scale to the extreme limit of our computing capability. The width of graphene sheet was increased from 22 nm to 200 nm and 440 nm. The total number of C atoms in the 440 nm wide graphene sheet was 2,985,734, which was 400 times larger than the number of atoms in our previous calculation. To simulate such large system for 5 ns, each MD task required 52,560 CPU hours, which corresponded to a non-stop calculation time of 20 days on 120 CPUs. As shown in Fig. R2 in the response to referee #1, the MD results demonstrate that the thermal rectification ratio decreases with increasing width, following a logarithmic curve ($R^2 = 0.989$). The MD simulation results agree well with the experimental data of sample #5, proving the validity of our result and analysis.

2. Possible mechanisms for thermal rectification in graphene

For the samples #4 and #5, the MD simulation result indicates that the asymmetric phonon localization is the reason for thermal rectification. In a trapezoid graphene ribbon, the narrow end causes stronger phonon lateral confinement and produces more

localized phonon modes as collision centers ^[2]. Hence, the narrow end of graphene ribbon becomes the bottleneck for the delocalized phonons to travel in the direction from the narrow end to the wide end. The opposite direction from the wide end to the narrow end is the favored direction for thermal transport.

Actually, as a potential mechanism for thermal rectification, the phonon localization has already been proved by the other researchers in the MD simulation ^[2, 13, 14]. Besides this, some other possible mechanisms were also proposed to explain the thermal rectification behavior ^[1, 2, 4, 6, 8]: (1) Inseparable dependence of thermal conductivity on space and temperature. This is the mechanism confirmed in the graphene samples #1, #2, #3 (see Fig. 4 in the article). The asymmetric phonon-defect scattering plays a dominant role. (2) Phonon spectra overlap. The amount of phonon spectra overlap is different before and after reversing the heat flux direction, leading to thermal rectification. (3) Phonon spectra mismatch across the interface. The phonon spectra are different in two segments of the material. The thermal rectification can be interpreted as the different phonon spectra mismatch before and after reversing the heat flux direction. In this work, we have carefully checked all the possible mechanisms for the samples #4 and #5. There was no significant difference found in the temperature dependence of thermal conductivity or in the phonon spectra overlap and mismatch. The phonon localization effect was found to be the most explicit and promising reason for the thermal rectification.

3. Major contribution of current work

We would like to highlight that the most important contribution of current work is to provide the first experimental evidence for two different kinds of graphene thermal

rectifiers by using the state-of-the-art technology. In order to provide a reasonable explanation for the samples #4 and #5, we have spent a lot of time and effort in MD simulation to prove that the thermal rectification exists at different scales from nanometer to micrometer. Although the simulation model is still smaller than the experimental sample due to our limited computational resources and revision time, it should not affect the validity and novelty of the experimental contribution.

2. The uncertainty analysis is still not clear. I do not see how the non-uniform temperature in the heater/sensor is included in this analysis. I only see a discussion of the average temperature rise and measurement resolution. Also, what is the confidence of the uncertainty bands? 67%, 95%? Even if this is a 95% confidence interval, the data presented in Figure 6 suggests the rectifying effect is basically not present in samples #4 or #5. In this case, the data should only be included to suggest that geometry, at least at these length scales, and mass loading do not contribute to thermal rectification.

Response: Thank you for the important question about the measurement uncertainty. First, the non-uniform temperature distribution in the heater/sensor has been taken into account in this work. Fig. R5 shows one example of two-dimensional thermal analysis by COMSOL Multiphysics™.

Fig. R5 Temperature distribution of graphene sample #1 without nanopores. The left and right figures represent the different temperature distributions in two heat flux directions.

It is clear that the temperature is not uniform in the nanofilm heater marked by the black circle in the figure. In fact, the temperature of the sensor as thermometer is not uniform either, but the temperature distribution is not so obvious due to its small temperature rise. The average temperature rises of both heater and thermometer were directly calculated from the two-dimensional thermal analysis result.

Second, the confidence of the uncertainty bands comes from the detailed uncertainty analysis. As discussed in the supplementary material, the measurement uncertainties from different resources were estimated in this work, including the temperature of sensor, temperature fluctuation of Peltier stage, geometric dimensions of graphene device, current and voltage measurements and numerical thermal analysis. It was found that the largest uncertainty came from the temperature of sensor, which was $\sim 2.3\%$. Considering all these factors, the final uncertainty of thermal conductivity measurement was well below 5%. The uncertainty bands illustrated in Fig. 6 of the article were drawn based on the maximum value of 5%.

Here, 5% is the upper limit of the estimated uncertainty. More importantly, as illustrated in Fig. 6 of the article, the measurement has been repeated 6 times for both

samples #4 and #5 at different substrate temperatures, and the measured thermal conductivity was always higher in one specific heat flux direction. Although the exact value of 10% rectification ratio may be altered by some random error, the existence of thermal rectification in the samples #4 and #5 should be affirmative.

Another confidence for the high experimental accuracy comes from the high thermal sensitivity of our H-type sensor.

Fig. R6 Comparison between our H-type sensor and commonly used micro thermal bridge device [21].

Figure R6 shows the comparison between our H-type sensor and a micro thermal bridge device, which was used to measure the first thermal rectifier made from carbon nanotube with 7% efficiency. In the thermal bridge device, Pt film resistor was deposited on two suspended SiN_x pads with thickness of hundreds of nanometers. Each SiN_x pad was supported by several thin and long SiN_x ribbons above the substrate. The size of SiN_x pad was approximately 20 μm × 20 μm. In contrast, the size of our nanofilm sensor is much smaller, around 1 μm × 12 μm. More importantly, in our method, 100 nm thick Au sensor is suspended without any supporting substrate, much thinner than the SiN_x pad with Pt resistor. Consequently, our H-type sensor is expected to have much smaller thermal capacity and higher thermal sensitivity than the micro thermal bridge

device. It was claimed in the Ref. [21] that the measurement uncertainty of thermal bridge device was only 1%. So the 5% measurement uncertainty of our H-type sensor sounds quite reasonable.

Lists of changes

Here we summarize all the revisions made in the manuscript. Text revision is highlighted by red color in the manuscript.

1. We have added discussion on page 15 to highlight the different physical mechanisms of the samples #1, #2, #3 and samples #4, #5.
2. We have extended the domain size in MD simulation and plotted the thermal rectification ratio versus the graphene width in the Fig. 7b on page 16.
3. We have added discussion on page 17 to describe the dramatically increasing computational complexity with the increasing size of model.
4. We have added discussion on page 18 to explain how the asymmetric phonon localization causes thermal rectification in sample #5. A phonon MFP analysis is used to explain the thermal rectification in micrometer-sized graphene sample.
5. We have added discussion on page 20 to explain the physical mechanism of thermal rectification in sample #4.
6. We have updated the citation detail for Ref. 27.
7. In the supplementary material, we explained the computational complexity of large MD simulation model on page 25.
8. In the supplementary material, we explained the complicated lattice dynamics calculation on page 28.

References:

- [1] J. N. Hu, X. L. Ruan, Y. P. Chen, Thermal conductivity and thermal rectification in graphene nanoribbons: a molecular dynamics study, *Nano Lett.* 2009, 9, 2730-2735.
- [2] Y. Wang, A. Vallabhaneni, J. N. Hu, B. Qiu, Y. P. Chen, X. L. Ruan, Phonon lateral confinement enables thermal rectification in asymmetric single-material nanostructures, *Nano Lett.* 2014, 14, 592-596.
- [3] W. W. Zhao, Y. L. Wang, Z. T. We, et al., Defect-engineered heat transport in graphene: a route to high efficient thermal rectification, *Sci. Rep.*, 2015, 5, 11962.
- [4] N. Yang, G. Zhang, B. W. Li, Thermal rectification in asymmetric graphene ribbons, *Appl. Phys. Lett.*, 2009, 95, 033107.
- [5] Y. Wang, S. Chen, X. L. Ruan, Tunable thermal rectification in graphene nanoribbons through defect engineering: A molecular dynamics study, *Appl. Phys. Lett.*, 2012, 100, 163101.
- [6] C. Melis, G. Barbarino, L. Colombo, Exploiting hydrogenation for thermal rectification in graphene nanoribbons, *Phys. Rev. B*, 2015, 92, 245408.
- [7] X. J. Liu, G. Zhang, Y. W. Zhang, Graphene-based thermal modulator, *Nano Research*, 2015, 2, 0782.
- [8] W. R. Zhong, W. H. Huang, X. R. Deng, B. Q. Ai, Thermal rectification in thickness-asymmetric graphene nanoribbons, *Appl. Phys. Lett.*, 2011, 99, 193104.
- [9] Q. Liang, Y. Wei, Molecular dynamics study on the thermal conductivity and thermal rectification in graphene with geometric variations of doped boron, *Physica B*, 2014, 437, 36-40.

- [10] H. Y. Cao, H. J. Xiang, X. G. Gong, Unexpected large thermal rectification in asymmetric grain boundary of graphene, *Solid State Communications*, 2012, 152, 1807-1810.
- [11] H. B. Fan, L. Deng, X. M. Yuan, J. Guo, X. L. Li, P. Yang, Thermal conductivity and thermal rectification in H-terminated graphene nanoribbons, *RSC Adv.*, 2015, 5, 38001-38005.
- [12] P. Yang, X. L. Li, H. Y. Yang, X. N. Wang, Y. Q. Tang, X. M. Yuan, Numerical investigation on thermal conductivity and thermal rectification in graphene through nitrogen-doping engineering, *Appl. Phys. A*, 2013, 112, 759-765.
- [13] J. H. Lee, V. Varshney, A. K. Roy, J. B. Ferguson, B. L. Farmer, Thermal rectification in three-dimensional asymmetric nanostructure, *Nano Lett.*, 2012, 12, 3491-3496.
- [14] N. Yang, G. Zhang, B. W. Li, Carbon nanocone: A promising thermal rectifier, *Appl. Phys. Lett.*, 2008, 93, 243111.
- [15] S. Ghosh, I. Calizo, D. Teweldebrhan, E. P. Pokatilov, D. L. Nika, A. A. Balandin, W. Bao, F. Miao, C. N. Lau, Extremely high thermal conductivity of graphene: Prospects for thermal management applications in nanoelectronic circuits, *Appl. Phys. Lett.*, 2008, 92, 151911.
- [16] G. Fugallo, A. Cepellotti, L. Paulatto, M. Lazzeri, N. Marzari, F. Mauri, Thermal conductivity of graphene and graphite: collective excitations and mean free paths, *Nano Lett.*, 2014, 14, 6109-6114.
- [17] X. F. Xu, L. F. C. Pereira, Y. Wang, J. Wu, K. W. Zhang, X. M. Zhao, S. Bae, C. T. Bui, R. G. Xie, J. T. L. Thong, B. H. Hong, K. P. Loh, D. Donadio, B. W. Li, B. Ozyilmaz, Length-dependent thermal conductivity in suspended single-layer

- graphene, *Nature Commun.*, 2014, 5, 3689.
- [18] H. D. Wang, K. Kurata, T. Fukunaga, X. Zhang, H. Takamatsu, Width dependent intrinsic thermal conductivity of suspended monolayer graphene, *Int. J. Heat Mass Transfer*, 2017, 105, 76-80.
- [19] S. Ghosh, I. Calizo, D. Teweldebrhan, E. P. Pokatilov, D. L. Nika, A. A. Balandin, W. Bao, F. Miao, C. N. Lau, Extremely high thermal conductivity of graphene: Prospects for thermal management applications in nanoelectronic circuits, *Appl. Phys. Lett.*, 2008, 92, 151911.
- [20] G. Fugallo, A. Cepellotti, L. Paulatto, M. Lazzeri, N. Marzari, F. Mauri, Thermal conductivity of graphene and graphite: collective excitations and mean free paths, *Nano Lett.*, 2014, 14, 6109-6114.
- [21] C. W. Chang, D. Okawa, A. Majumdar, A. Zettl, Solid-state thermal rectifier, *Science*, 2006, 314, 1121-1124.

Reviewers' Comments:

Reviewer #1:

Remarks to the Author:

I am troubled by the use of the term phonon localization in the manuscript for explaining the results of samples #4 and 5. When the graphene width is smaller than the phonon mean free path, phonon transport is in the ballistic or Casimir regime instead of the localization regime. Strongly localized vibration modes do not make direct contribution to heat conduction, and have to rely on anharmonic coupling with propagating modes to conduct heat. In comparison, weakly localized modes can contribute to heat conduction via diffusive random walks. A good discussion of phonon localization can be found in Allen and Feldman, *Physical Review B* 48, 12581 (1993). Localized and diffusive modes are usually found in highly disorder or amorphous structures, where few atoms participate in a localized or diffusive vibration mode. In a perfect graphene crystal with a finite size as large as ~ 0.5 micron, it is hard to imagine that few of the many atoms within the still large width participate in the vibration mode in order for the mode to be localized, especially given the large mean free path for phonon-phonon scattering in high quality graphene. The effective mean free path is in the atomic scale for a diffusive mode and vanishes for a strongly localized mode.

The calculated participation ratio at the hot side is lower when hot side is the narrower end than when the hot side is the wider side. However, this result is insufficient to suggest that phonon localization has occurred. Instead, it simply reveals shorter mean free paths of propagating modes due to edge scattering at the narrower ends.

The most intriguing feature of the calculation results is that the calculated participation ratio at the cold side is lower when the cold side is the wider end than when the cold side is the narrower end. The authors have suggested that this feature is associated with the lower participation ratio calculated at the hot side when the hot side is the narrower end. Such a connection could be possible. The phonon population density is supposed to be relatively high at the hot side compared to the cold side. When the hot side is the narrower end, the mean free paths of the large-population phonons at the hot side can be reduced more, so that fewer propagating phonon modes can reach the cold side. In other words, the edge scattering effect affects a larger phonon population when the hot side is the narrower end. However, this effect should not be called localization, because the vibration modes are still propagating modes with mean free paths as long as the ~ 0.5 micron ribbon width.

Moreover, the participation ratio of the cold side can be affected by that of the hot side because the phonon-phonon scattering mean free paths of some vibration modes can be as large as the length of the graphene ribbon. If the phonon-phonon scattering mean free path is much shorter than the ribbon length, the participation ratios at the two ends should be independent to each other. These modes with long phonon-phonon scattering mean free path would be in the ballistic transport regime. For ballistic transport along the ribbon length, there would be large temperature drops at the two contacts, which give rise to large contact thermal resistance. As such, the observed rectification for high-quality graphene ribbon is expected to be influenced by temperature and direction dependences of the contact thermal resistance.

In summary, the experimental results appeared to be interesting. However, the explanation based on phonon localization is incorrect, although several recent papers might have also incorrectly used the term localization for describing boundary scattering of propagating modes. In my opinion, the paper can be published only after the problematic conceptual issues in the manuscript are corrected and replaced with a meaningful explanation of the experimental observation for samples #4 and 5, and after the contact thermal resistance issue and its influence on the observed thermal rectification is adequately addressed.

Reviewer #2:

Remarks to the Author:

I am satisfied with the additional responses by the authors and feel this manuscript should be published in *Nature Communications*.

Point-to-point response

Manuscript No. NCOMMS-16-24614A-Z

Title: Experimental study of thermal rectification in suspended monolayer graphene

Reviewers' comments:

Reviewer #1 (Remarks to the Author):

I am troubled by the use of the term phonon localization in the manuscript for explaining the results of samples #4 and 5. When the graphene width is smaller than the phonon mean free path, phonon transport is in the ballistic or Casimir regime instead of the localization regime. Strongly localized vibration modes do not make direct contribution to heat conduction, and have to rely on anharmonic coupling with propagating modes to conduct heat. In comparison, weakly localized modes can contribute to heat conduction via diffusive random walks. A good discussion of phonon localization can be found in Allen and Feldman, Physical Review B 48, 12581 (1993). Localized and diffusive modes are usually found in highly disorder or amorphous structures, where few atoms participate in a localized or diffusive vibration mode. In a perfect graphene crystal with a finite size as large as ~ 0.5 micron, it is hard to image that few of the many atoms within the still large width participate in the vibration mode in order for the mode to be localized, especially given the large mean free path for phonon-phonon scattering in high quality graphene. The effective mean free path is in the atomic scale for a diffusive mode and vanishes for a strongly localized mode.

The calculated participation ratio at the hot side is lower when hot side is the

narrower end than when the hot side is the wider side. However, this result is insufficient to suggest that phonon localization has occurred. Instead, it simply reveals shorter mean free paths of propagating modes due to edge scattering at the narrower ends.

The most intriguing feature of the calculation results is that the calculated participation ratio at the cold side is lower when the cold side is the wider end than when the cold side is the narrower end. The authors have suggested that this feature is associated with the lower participation ratio calculated at the hot side when the hot side is the narrower end. Such a connection could be possible. The phonon population density is supposed to be relatively high at the hot side compared to the cold side. When the hot side is the narrower end, the mean free paths of the large-population phonons at the hot side can be reduced more, so that fewer propagating phonon modes can reach the cold side. In other words, the edge scattering effect affects a larger phonon population when the hot side is the narrower end. However, this effect should not be called localization, because the vibration modes are still propagating modes with mean free paths as long as the ~ 0.5 micron ribbon width.

Moreover, the participation ratio of the cold side can be affected by that of the hot side because the phonon-phonon scattering mean free paths of some vibration modes can be as large as the length of the graphene ribbon. If the phonon-phonon scattering mean free path is much shorter than the ribbon length, the participation ratios at the two ends should be independent to each other. These modes with long phonon-phonon scattering mean free path would be in the ballistic transport regime. For ballistic transport along the ribbon length, there would be large temperature drops at the two contacts, which give rise to large contact thermal resistance. As such, the observed

rectification for high-quality graphene ribbon is expected to be influenced by temperature and direction dependences of the contact thermal resistance.

In summary, the experimental results appeared to be interesting. However, the explanation based on phonon localization is incorrect, although several recent papers might have also incorrectly used the term localization for describing boundary scattering of propagating modes. In my opinion, the paper can be published only after the problematic conceptual issues in the manuscript are corrected and replaced with a meaningful explanation of the experimental observation for samples #4 and 5, and after the contact thermal resistance issue and its influence on the observed thermal rectification is adequately addressed.

Response:

We deeply appreciate the reviewer's careful examination of our manuscript and the constructive comments. We also appreciate reviewer's clarification of the concept of phonon localization. The idea of using phonon localization to explain the thermal rectification phenomena came from Ruan's paper "*Phonon lateral confinement enables thermal rectification in asymmetric single-material nanostructures*" [*Nano Letters* 14, 592-596 (2014)], where the author used the concept of phonon edge localization to explain the thermal rectification in T-shaped graphene nanoribbon. As clarified by the reviewer, we have realized that the concept of phonon localization might not be appropriate for graphene, since the localized phonon modes usually exist in highly disorder or amorphous structures with very short mean free paths comparable to the interatomic spacing ^[1]. Following the reviewer's suggestion, we have removed the discussion of phonon localization and renewed the physical explanation in terms of

asymmetric edge scattering for the samples #4 and #5.

Fig. R1 Distribution of the spatial energy for the propagating phonon modes in two opposite heat flux directions of the trapezoid graphene (redrawn from Fig. 7 in the article). T_h and T_c are the high and low temperatures at two ends of graphene. (c) shows the spatial energy distribution for the heat flux from the wide end to the narrow end. (d) shows the spatial energy distribution from the narrow end to the wide end. The local energy E of propagating phonon modes is always smaller in case (d) than that in case (c) at both high-temperature and low-temperature ends.

In order to explain the physical mechanism of thermal rectification more clearly, Fig. 7 (c-d) from the main article is redrawn here as Fig. R1. The lattice dynamics calculation result indicates that the local energy E of propagating phonon modes is always smaller in the heat flux direction from the narrow end to the wide end. As inspired by the reviewer, we would like to discuss the spatial energy distribution separately for the high and low temperatures. (In the last revised manuscript, Figs. 7 and 8 in the manuscript are the distributions of spatial energy of propagating modes, not the distribution of participation ratio. We included the propagating phonon modes with the participation ratio larger than 0.4 into the spatial energy calculation. The mathematical

definitions of P_λ and E can be found in the main text.) At the high temperature T_h , E at the narrow end of graphene is smaller than that at the wide end, i.e. $E_4 < E_1$. It reveals that the narrow end of graphene has stronger edge scattering effect on the propagating phonon modes than the wide end. Hence, the phonon mean free path and the local energy of propagating modes are smaller at the narrow end.

At the low temperature T_c , E at the wide end of graphene is smaller than that at the narrow end, i.e. $E_3 < E_2$. This feature is associated with the low energy of propagating phonon modes at the narrow end of graphene under the high temperature condition. The phonon population density will be higher at the hot side of graphene than that at the cold side. If the hot side is the narrow end, the mean free paths of the large-population phonons will be reduced more. As a result, fewer phonon modes at the hot side can propagate to the cold side, so that the local energy of propagating modes at the cold side (wide end) is relatively small. In other words, the narrow end of graphene at high temperature appears to be the “bottleneck” for the propagating phonon modes. In this way, the thermal conductivity in the heat flux direction from the narrow end to the wide end is smaller than that in the opposite direction. Similar to the case of tapered width in sample #5, depositing carbon nanoparticles at one side of sample #4 can also cause asymmetric phonon scattering for the propagating modes.

Fig. R2 Distribution of the spatial energy for the propagating phonon modes in two opposite heat flux directions of the asymmetrically deposited graphene (redrawn from Fig. 8 in the article). T_h and T_c are the high and low temperatures at two ends of graphene. (b) shows the energy distribution for the heat flux from the deposited region to the clean region. (c) shows the energy distribution from the clean region to the deposited region. The local energy E of propagating modes is always smaller in case (b) than that in case (c) at both high-temperature and low-temperature ends.

Figure R2 is redrawn from Fig. 8 in the article to explain the physical mechanisms of thermal rectification. The asymmetric distribution of the spatial energy for the propagating phonon modes in Fig. R2 is quite similar to the result of Fig. R1. The deposited heavy atoms cause significant phonon scattering in the graphene sheet, leading to the reduced local energy of propagating modes in the heat flux direction from the deposited region to the clean region ($E_4 < E_1, E_3 < E_2$). The deposited region at high temperature appears to be the “bottleneck” for the propagating phonon modes in graphene, so that fewer propagating modes can reach the cold side. Hence, the energy distribution of propagating phonon modes and thermal conductivity are smaller in the

case of Fig. R2b.

Figure 7 (b) in the article indicates that the thermal rectification ratio decreases as the geometric size of graphene increases. The molecular dynamics simulation results agree well with the experimental data. The phonon-phonon scattering mean free path in graphene is close to or even larger than the size of sample ^[2]. In this case, the local energies of propagating modes at the two ends of graphene are coupled to each other. The propagating phonon modes at the cold side are affected by the phonon scattering at the hot side. If the length or width of graphene is much larger than the phonon mean free path, the local energies at the hot and cold sides are independent to each other, then the thermal rectification disappears.

The above discussion of thermal rectification has been added on pages 17, 18 and 20 of the revised manuscript and marked in red color.

Effect of contact thermal resistance

The reviewer mentioned that the thermal rectification might be affected by the contact thermal resistance. In order to give a satisfactory response to this issue, we carefully examined the contact thermal resistance between the graphene and metallic sensor. The contact thermal resistance R_c was calculated by using a standard fin thermal resistance model ^[3]:

$$R_c = \left[\sqrt{\frac{\lambda AW}{R_{int}}} \tanh \left(\sqrt{\frac{W}{\lambda AR_{int}}} L_c \right) \right]^{-1}, \quad (\text{R1})$$

where $R_{int} = 4 \times 10^{-8} \text{ m}^2\text{K/W}$ is the interfacial thermal resistance per unit area ^[4], λ and W are the thermal conductivity and width of the graphene sample, $A = dW$ is the cross-sectional area (d is the thickness of monolayer graphene), L_c is the graphene-metal

contact length. As summarized in Table 1 of the supplementary material, the contact thermal resistance is about 10% of the thermal resistance of monolayer graphene.

For the rectangle graphene ribbons (#1 to #4), the contact thermal resistances at the two ends of graphene are the same because of the uniform width. Hence, the contact thermal resistance has no influence on the thermal rectification. We have measured the thermal conductivities of pristine graphene in two opposite heat flux directions before the defect engineering and carbon deposition. No thermal rectification was found in these rectangle graphene ribbons. More importantly, the graphene ribbon was supported below the metallic sensor at the contact points. The 100nm thick gold film protected the graphene lattice from the ion beam radiation or the electron beam induced deposition. Hence, the contact thermal resistance was unchanged before and after defect engineering, and it has no contribution to the measured thermal rectification from sample #1 to #4.

For the trapezoid graphene ribbon #5, the contact thermal resistances are different at two ends. According to the Eq. (R1), the wide end of graphene has smaller contact thermal resistance. The ratio between two contact thermal resistances is calculated as $R_{c-wide}/R_{c-narrow} \sim 0.5$. It is noted that two contact thermal resistances and the thermal resistance of graphene ribbon are connected in series. The measured thermal rectification is related to the different total thermal resistances in two opposite heat flux directions. Assuming that the thermal rectification is only caused by the different contact thermal resistances at two ends, the maximum value can be calculated as:

$$\varphi_{max} = \frac{\Delta R}{R_0 + 2R_{c-narrow}} = \frac{2(R_{c-narrow} - R_{c-wide})}{R_0 + 2R_{c-narrow}} \approx 11\% , \quad (R2)$$

where R_0 is the thermal resistance of suspended graphene ribbon. This maximum value

is the same as the measured rectification ratio of sample #5 in the experiment. However, the real rectification ratio caused by the asymmetric contact thermal resistance should be much smaller than φ_{max} , because the different temperature dependence of contact thermal resistances is essential for rectification. If the contact thermal resistance is independent of temperature, the asymmetric contact thermal resistance has no effect on the thermal rectification. According to the Eq. (R1), the temperature dependence of contact thermal resistance may come from the thermal conductivity of supported graphene λ and the interfacial thermal resistance R_{int} . Shi et al. measured the thermal conductivity of supported monolayer graphene on SiO₂ and the result was almost constant in the temperature range from 300K to 400K [5]. Pop et al. measured the interfacial thermal conductance of Au/Ti/graphene/SiO₂ structure (similar to our Au/Cr/graphene structure) by using a time-domain thermoreflectance method from 50K to 500K [4]. The result indicates that the interfacial thermal conductance is almost constant from 300K to 500K. The present experimental results demonstrate that λ and R_{int} are almost independent of temperature above 300K. Hence, the contact thermal resistance is also independent of temperature and has no contribution to the thermal rectification.

From the microscopic point of view, the phonon-substrate scattering dominates in the supported graphene. So the thermal conductivity of supported graphene is significantly lower than that of the suspended graphene, and independent of temperature from 300K to 400K. In contrast, the thermal conductivities of samples #4 and #5 decrease with increasing temperature, where phonon-phonon scattering plays an important role.

The above discussion of contact thermal resistance has been added on page 12 of the revised supplementary material and marked in red color.

Reference:

- [1] P. B. Allen, J. L. Feldman, Thermal conductivity of disordered harmonic solids, *Phys. Rev. B* 48 (1993) 581-588.
- [2] G. Fugallo, A. Cepellotti, L. Paulatto, M. Lazzeri, N. Marzari, F. Mauri, Thermal conductivity of graphene and graphite: collective excitations and mean free paths, *Nano Lett.*, 14 (2014) 6109-6114.
- [3] M. T. Pettes, I. Jo, Z. Yao, L. Shi, Influence of polymeric residue on the thermal conductivity of suspended bilayer graphene, *Nano Lett.* 11 (2011) 1195-1200.
- [4] Y. K. Koh, M. H. Bae, D. G. Cahill, E. Pop, Heat conduction across monolayer and few-layer graphenes, *Nano Lett.* 10 (2010) 4363-4368.
- [5] J. H. Seol, I. Jo, A. L. Moore, L. Lindsay, Z. H. Aitken, M. T. Pettes, X. S. Li, Z. Yao, R. Huang, D. Broido, N. Mingo, R. S. Ruoff, L. Shi, Two-dimensional phonon transport in supported graphene, *Science* 328 (2010) 213-216.

Reviewer #2 (Remarks to the Author):

I am satisfied with the additional responses by the authors and feel this manuscript should be published in Nature Communications.

Reviewers' Comments:

Reviewer #1:

Remarks to the Author:

I am satisfied with the revised version, and I agree that publication of this version of the manuscript can be of interest for the broad readership of Nature Communications.